# Simulated density reorganization on the Weddell Sea continental shelf sensitive to atmospheric forcing

Vanessa Teske[1,2], Ralph Timmermann[1], Cara Nissen[3,4], Rolf Zentek[5,6], Tido Semmler[1,7], and Günther Heinemann[5]

[1]Department of Biogeochemical Modelling, GEOMAR, D-24148 Kiel, Germany
[2]Alfred Wegener Institute for Polar and Marine Research, D-27570 Bremerhaven, Germany
[3]Department of Atmospheric and Oceanic Sciences and Institute of Arctic and Alpine Research, University of Colorado, Boulder, Boulder, Colorado, USA
[4]Department of Freshwater and Marine Ecology, Institute for Biodiversity and Ecosystem Dynamics, University of Amsterdam, Netherlands
[5]Department of Environmental Meteorology, University of Trier, D-54286 Trier, Germany
[6]German Weather Service , D-63067 Offenbach, Germany
[7]Met Éireann, 65-67 Glasnevin Hill, D09 Y921 Dublin, Ireland

**Correspondence:** Vanessa Teske (vteske@geomar.de)

**Abstract.**

The strong Antarctic Slope Front in the southern Weddell Sea limits the present-day transport of modified Warm Deep Water onto the continental shelf and is associated with a characteristic V-shape in the density structure across the continental slope. The mechanisms controlling today's V-shape are well studied, but its future development is not yet well constrained. In this study, we run ocean model simulations for a 21st-century Shared Socio-economic Pathways (SSP) 3-7.0 emission scenario. The forcing is retrieved from atmospheric model output from simulations with a global climate model and from a higher-resolved regional atmospheric model, respectively. We find that the resolution of the atmospheric model component influences the simulated future transport of modified Warm Deep Water onto the continental shelf into the Filchner Trough in the southern Weddell Sea through differences in the evolution of the depth and symmetry of the V-shape over the 21st century. In both simulations, reduced sea-ice formation and weakened Ekman downwelling reduce the depth of the V-shape and increase the sensitivity of its position above the slope to seasonal variations in sea-ice production and in the wind field. Using forcing data from an atmosphere model with higher resolution leads to an acceleration of the density redistribution on the continental shelf compared to the simulations forced with coarse resolution data. This indicates that the SSP3-7.0 climate scenario may have a higher potential for a regime shift from cold to warm Filchner Trough through a cross-slope current before the end of the 21st century than other ocean simulations for the same scenario but with lower atmospheric resolution suggest. As cross-slope currents disturb the continuity of the V-shape, we define a spatial grade of connectivity to quantify the lateral integrity of the V-shape along the continental slope. We find that the integrity of the V-shape reduces with a delay of 3 months after a strong cross-slope current of modified Warm Deep Water enters Filchner Trough. Atmospheric downscaling increases the potential for a regime shift, dominated by warmer summer air temperatures. The Antarctic Slope Front is temporarily disturbed by cross-slope currents but the primary reason for the regime shift is the cross-slope density gradient.

# 1 Introduction

The Filchner Trough on the continental shelf in the southern Weddell Sea, Antarctica (Fig. 1), is one of the key regions of Dense Shelf Water (DSW) export and therefore plays an important role in the global ocean circulation (Heywood et al., 2014). However, the Filchner Trough is also a region where an onshore current of modified Warm Deep Water (mWDW), the local cooler derivative of Circumpolar Deep Water, may reach the Filchner ice shelf cavity and significantly increase melt rates (Darelius et al., 2016; Ryan et al., 2020). Future climate projections for different warming scenarios have shown that a regime shift from a cold DSW-dominated to a warm mWDW-dominated Filchner Trough is possible during the 21st century (Timmermann and Hellmer, 2013; Daae et al., 2020; Haid et al., 2023; Nissen et al., 2023; Teske et al., 2024). Numerous studies have demonstrated that the density ratio between the continental shelf and the open ocean is critical in controlling the on-shelf transport of relatively warm off-shore water (Daae et al., 2020; Haid et al., 2023; Nissen et al., 2023). While a higher density of shelf waters prevents a warm inflow onto the shelf, projections have shown the potential for a reversal of the density ratio (Haid et al., 2023; Nissen et al., 2023), allowing for mWDW transport onto the continental shelf.

North of Filchner Trough above or near the continental shelf break sits the Antarctic Slope Front (ASF), the frontal zone between coastal and open-ocean waters which is associated with strong subsurface temperature and salinity gradients. Following the ASF, a coherent westward circulation transports mWDW into the southern Weddell Sea. This current encircling Antarctica is the Antarctic Slope Current (ASC) (Thompson et al., 2018). Around Antarctica, the structure of the ASF varies depending on hydrographic properties over the continental shelf and shelf break, and can be classified into three groups: fresh shelf, warm shelf, and dense shelf (Thompson et al., 2018). With its DSW production and export, the continental shelf of the southern Weddell Sea is an example for a dense shelf. For dense shelves, density surfaces tilt down onshore towards the continental slope, but shoal again above the DSW layer. This creates a characteristic V-shape in the isopycnals running perpendicular to the continental slope (see also Fig. 2a) which has been described in many studies (e.g., Gill, 1973; Ou, 2007; Baines, 2009; Thompson et al., 2018).

Temporal variability of the V-shape has been linked to variability in both DSW export and on-shelf transport of mWDW. While the northern arm of the V-shape is formed by Ekman downwelling (Sverdrup, 1954), the southern arm of the V-shape is formed by entrainment of overlying surface water to the descending flow of DSW (Gill, 1973; Baines, 2009). The V-shape has been shown to be sensitive to the wind field(Graham et al., 2013). However, existing studies do not all agree on the importance of winds. A peak in wind strength in the southeastern Indian Ocean in April 2009 led to increased mixing, on-shore Ekman transport and convergent downwelling on the continental shelf near 113°E. Similar behaviour was observed in 2010 and 2011 (Peña-Molino et al., 2016). Kida (2011) showed in a numerical model experiment with an idealized set-up of the Filchner Trough and the southern Weddell Sea that enhanced winds lead to a deepening of the V-shape and a decrease of the ocean stratification near the continental shelf break. The larger amount of lighter surface water at greater depth increases the density gradient across the shelf break and enhances the geostrophically controlled overflow transport. In contrast, three-dimensional eddy-resolving simulations by Stewart & Thompson (Stewart and Thompson, 2015) showed only low sensitivity of the DSW export to wind strength. Additionally, the V-shape shows seasonal variability in depth that has been associated with variations in

the along-shore wind strength (Graham et al., 2013). Reanalysis of ship-based observations, Argo floats and data from marine mammals showed a steepening of the angle of the northern arm of the V-shape in the southern Weddell Sea in winter and a flattening in summer (Pauthenet et al., 2021; Le Paih et al., 2020). The seasonal variability of the V-shape and the associated thermocline at the continental slope leads to seasonal pulses of mWDW flowing into the Filchner Trough (Årthun et al., 2012; Hellmer et al., 2017). Several studies have shown that an intensification of these seasonal pulses is a precursor for an impending regime shift in the Filchner Trough (Hellmer et al., 2017; Naughten et al., 2021; Teske et al., 2024).

Despite its importance for the on-shelf supply of heat governing Antarctic ice shelf melting, only comparatively few studies with global ocean or climate models have focused on the ASC or the ASF due to the high resolution requirements to adequately resolve coastal ocean dynamics (Mathiot et al., 2011; Stewart et al., 2019; Beadling et al., 2022; Huneke et al., 2022). Additionally, on longer timescales, the ASF is remotely influenced by large-scale climate modes such as the Southern Annular Mode (SAM) and the El Niño-Southern Oscillation (ENSO) (Armitage et al., 2018; Spence et al., 2014, 2017). While Beadling et al. (2022) concentrated on possible responses of the ocean to changes in the wind field and meltwater input in a future climate, their model resolution of 100 km to 200 km is relatively coarse. Mathiot et al. (2011) showed that a downscaling of the atmospheric forcing to 40 km in a hindcast scenario increases katabatic winds and increases the strength of the seasonal cycle in the wind and temperature fields. While modelling studies on different aspects that affect the ASC, like freshwater input or changes in the wind strength and direction are more abundant, they often use idealized set-ups, regional models, or are only coarsely resolved (Kida, 2011; Nøst et al., 2011; Dinniman et al., 2012; St-Laurent et al., 2013; Hattermann et al., 2014; Lockwood et al., 2021; Ong et al., 2023). Previous studies have described a possible regime shift in the Filchner Trough from a DSW-dominated trough circulation to a mWDW-dominated circulation in the trough and possible consequences, including flooding of the ice shelf cavity with warm water from the deep ocean, a rise in basal melt rates of the ice shelves, reduced density of the exported shelf waters and a less efficient deep-ocean carbon and oxygen transfer (Hellmer et al., 2012; Timmermann and Hellmer, 2013; Naughten et al., 2021; Nissen et al., 2022, 2023, 2024). Less attention has been paid to the consequences that a shift in the current regime might have for the density structure at the continental slope which is controlling the mWDW inflow into Filchner Trough. The sensitivity of the ocean processes in response to the atmospheric forcing was demonstrated by Hattermann et al. (2014), Haid et al. (2015) and Dinniman et al. (2012). This leads us to the hypothesis that resolved mesoscale atmospheric processes may intensify the seasonality of the V-shape, the on-shore mWDW transport and the export of DSW in the Weddell Sea. Additionally, a finer atmospheric resolution produces more detailed and more pronounced temperature and wind speed gradients mostly related to katabatic winds and foehn wind (Van Lipzig et al., 2004; Mathiot et al., 2011; Van Wessem et al., 2015; Elvidge et al., 2014; Cape et al., 2015).

The aim of this study is to explore the evolution of the V-shaped ASF in the southwestern Weddell Sea in a warming climate (for study area see Fig. 1). By using ocean model simulations forced with data from two atmosphere models with different grid resolutions (see section 2), we assess the processes governing the structure of the ASF and the transport of mWDW onto the continental shelf. To reach that goal, we analyse (i) how the cross-slope structure of the V-shape at the Filchner Trough sill develops in a high-emission climate scenario for the 21st century in Sections 3.2 and 3.3, (ii) the change in seasonal atmospheric variability and its influence on the symmetry and structure of the V-shape (Sections 3.4 and 3.5), and (iii) the

longitudinal integrity of the V-shape as an indicator for the stability of the ASF and a possible regime shift in the Filchner Trough in a warming climate (Section 3.6). A brief description of the models and the methods used for analysis are given in Section 2, all results will be discussed in Section 4.

## 2  Methods

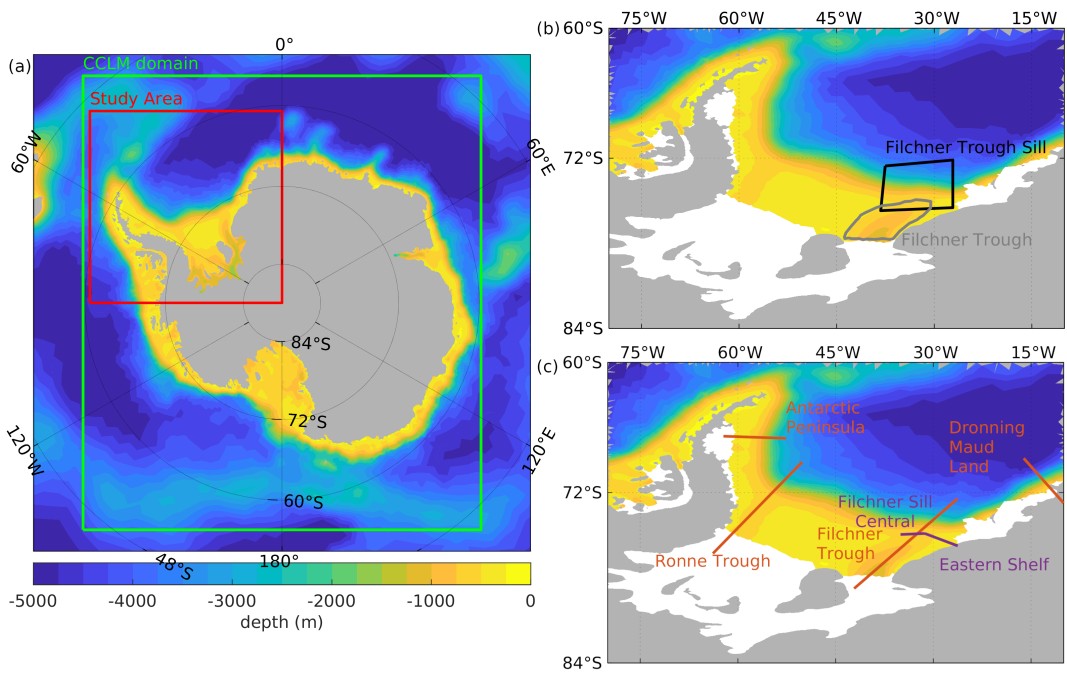

**Figure 1.** a) FESOM bottom topography in the Southern Ocean, study area in the Weddell Sea (red rectangle) and perimeter of the COSMO-Climate Model (CCLM) domain (green rectangle). b) Location of subregions and c) sections used in the analysis of this study. Areas shaded in white represent ice shelves.

### 2.1  The ocean model FESOM

For this study, we performed ocean model simulations with the Finite Element Sea-ice Ocean Model FESOM-1.4 (Wang et al., 2014). FESOM is a global ocean general circulation model with an unstructured mesh coupled with a dynamic-thermodynamic sea ice component, also including a representation of ice-shelf cavities (Timmermann et al., 2012; Wang et al., 2014; Danilov et al., 2015). The interaction between the ocean and ice shelves is governed by the three-equation parametrization that describes the flux of heat and fresh water through an exchange-controlling boundary layer (Hellmer and Olbers, 1989; Holland and

Jenkins, 1999). The variable horizontal resolution of the ocean mesh ranges from (minimum) 4 km around Antarctica and its adjacent ice shelf cavities, 25 km at 75°S in the open ocean in the Weddell Sea, to 250 km at the equator. In the vertical, the

mesh has 99 depth levels (z-levels) of increasing thickness with depth, spanning between 5 m near the surface, up to 337.5 m at nearly 6000 m depth (Gurses et al., 2019; Nissen et al., 2023). The model utilizes the ocean bathymetry, ice shelf geometry and grounding line position data set RTopo-2 (Schaffer et al., 2016). Parametrizations of subgrid-scale fluxes use the Gent and McWilliams (1990) scheme and Redi (1982) rotated tracer diffusion. Further detail on parameterizations in FESOM-1.4 has been provided by Wang et al. (2014). Bulk formulae for momentum transfer between the atmosphere and the ocean/sea-ice surface are quadratic functions of the velocity difference. The drag coefficient and the transfer coefficients for latent and sensible heat fluxes vary as a function of stability following Large and Yeager (2004). Here, we perform two ocean model simulations forced with atmospheric data from either a global climate model ("REF" simulation throughout the paper) or a higher-resolved regional atmospheric model (referred to as the "FECO" simulation) which will be described in more detail in the following section.

## 2.2 The atmospheric forcing data

The FESOM reference simulation REF is forced with global atmospheric data created with the AWI Climate Model (AWI-CM), a coupled global climate model using FESOM1.4 as the ocean model and ECHAM6.3.04p1 as the atmospheric model component. The data was created as a contribution to phase 6 of the Coupled Model Intercomparison Project (CMIP6; Semmler et al., 2020). The atmospheric data resolution ranges from approx. 27 km to 52 km in zonal direction and corresponds to approx. 100 km in meridional direction. Our ocean model simulation is initialized in the year 2000 with FESOM output from Nissen et al. (2023) and is driven by 3-hourly atmospheric output of the historical AWI-CM simulation for the period 2000 to 2014 and of the future Shared Socio-economic Pathways (SSP) 3-7.0 emission scenario (Meinshausen et al., 2020) projection for 2015 to 2100, identical to Nissen et al. (2023).

The FESOM FECO simulation is forced with a regional atmospheric forcing data set which was created with the COnsortium for Small-scale MOdeling - CLimate Mode model (COSMO-CLM or CCLM; Steger and Bucchignani, 2020; Zentek and Heinemann, 2020), a regional non-hydrostatic atmospheric model with terrain-following vertical coordinates at a horizontal resolution of 15 km. The used CCLM is a polar-adapted version including improved parameterizations for sea ice and the stable boundary layer (see Heinemann et al., 2022, for more details). The model is used in a forecast mode with daily re-initialization and a spin-up of 12 hours. The CCLM domain reaches the northernmost corners at approx. 50°S (Fig. 1a). At its lateral boundaries, it is driven by the global AWI-CM data sets described above. Sea ice concentration and thickness are initially taken from FESOM data of the AWI-CM simulation, but are modified by the parameterizations of grid-scale and sub-grid-scale ice in leads and polynyas. CCLM was evaluated for the present days climate for the Weddell Sea region in Zentek and Heinemann (2020) using near-surface data, upper-air data, ERA reanalyses (ERA-Interim and ERA5) and the Antarctic Mesoscale Prediction System (AMPS Powers et al., 2012) for the period 2002–2016. CCLM showed a good representation of temperature and wind for the Weddell Sea region. For use as forcing for the global ocean model FESOM, the CCLM output is merged with the global AWI-CM atmospheric output. The merging follows the procedure from Haid et al. (2015) and uses a transition zone of 2° to linearly interpolate between the global AWI-CM data and the regional CCLM data. South of this 2° boundary only CCLM data is applied as forcing, while only AWI-CM data is used outside the CCLM domain. Inside the

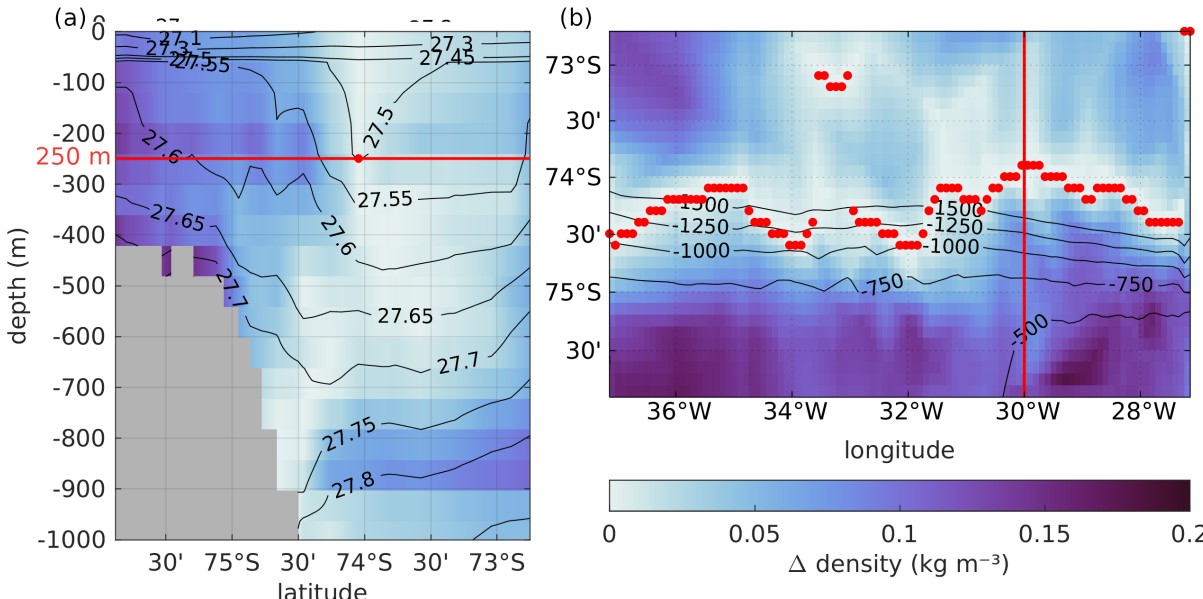

**Figure 2.** Section across Filchner Trough sill of the horizontal potential density distribution relative to the level-wise horizontal minimum density in December 2000 to illustrate the connection between the V-shaped density distribution and the grade of connectivity as defined in section 2.3. Isolines depict the potential density in kg m$^{-3}$ relative to 1000 kg m$^{-3}$. The red dot marks the location of the minimum density difference at 250 m depth, the horizontal line the position of the level presented in (b). (b) Horizontal projection of the density difference at 250 m depth with bottom isobaths. Red dots mark the location of the meridional minimum in longitudinal increments of 0.1°. The red line marks the location of the section shown in panel (a).

transition zone, the weighted average depending on the distance to the CCLM domain boundary is used. CCLM data are available for the three time slices 2000-2014, 2036-2050, and 2086-2100. We ran FESOM simulations forced with CCLM (henceforth called FECO) for each of the three time slices, each preceded by 10 years of spin-up, already driven by the CCLM data (2000-2009, 2036-2045, and 2086-2095, respectively). The spin-ups were branched off the REF simulation in 2000, 2036, and 2086, respectively. The following atmospheric variables are used to drive all simulations: 2 m-temperature, 10 m-wind, downward longwave and shortwave radiation at ocean surface, mean sea level pressure, 2 m specific humidity and total precipitation.

## 2.3 Grade of Connectivity

We define the grade of connectivity (GOC) as a metric describing the coherency of the characteristic V-shape of the cross-slope isopycnals along the continental slope in a selected sector (here the Filchner Trough Sill, see Fig 1b). Motivated by the link between the V-shape and DSW export from the Weddell Sea continental shelf (Gill, 1973; Baines, 2009; Nissen et al., 2022) and on-shelf mWDW transport (Nøst et al., 2011; Stewart and Thompson, 2015; Thompson et al., 2018), this metric quantifies the modification (or erosion) of the V-shape structure in response to cross-slope currents. The GOC is defined as the normalized

zonal mean of the north-south variations of the deepest point of the V-shape in a horizontal plane. To calculate the GOC, we first interpolate the model data from the unstructured grid to a regular grid with a resolution of 0.1° through linear interpolation between FESOM grid points. As a second step, we find the horizontal density minimum at the chosen depth of 250 m (in the example in Fig, 2a it is $27.5\,kg/m^3$). This is repeated for each meridional grid coordinate, creating a number of minima (red dots in Fig. 2b). The average GOC is then calculated from monthly mean model output as

$$GOC = \frac{1}{N} \sum_n \begin{cases} 1, & if\ \Delta y_n \le d_L \\ \frac{d_L}{\Delta y_n}, & if\ \Delta y_n > d_L \end{cases} \tag{1}$$

where $\Delta y_n$ denotes the absolute meridional distance from the neighbouring minima $y_n$, $N$ is the number of grid cells in zonal direction, $n$ is the index running through the grid cells in zonal direction, and the distance threshold $d_L = 0.2°$ is the maximum distance that counts two neighbouring density minima as connected. In summary, the GOC is defined as the normalized zonal mean of the north-south variations of the deepest point of the V-shape in a horizontal plane. We normalize by $n$ so that it gives a number between 0 and 1, with lower values indicating a less coherent V-shape along the continental slope and a value of 1 indicating a V-shape without disruptions in zonal direction. Because the algorithm only includes the relative distance between the density minima, the GOC is robust against horizontal displacement of the whole V-shaped density structure. As such, the GOC allows for a quantitative description of the slope front stability and a quantitative assessment of how this relates to, e.g., on-shelf transport of mWDW.

## 3 Results

### 3.1 Present-day V-shape in the Weddell Sea (REF simulation)

The reference simulation REF shows a distinct wedge structure of the isopycnals above the continental slope for the period 2010-2014, but details vary substantially across different sectors of the Weddell Sea (Fig. 3). In the following, the potential density is always given relative to 1000 kg m$^{-3}$. At the coast of Dronning Maud Land (DML) in the eastern Weddell Sea, a fresh-shelf region, the narrow continental shelf prevents the accumulation of large amounts of DSW. Cold, fresh surface water is pushed towards the coast and subducted, pressing down on the isopycnals of greater density (Fig. 3a). The planes of equal density dip towards the continental shelf and the coast, forming only the northern arm of a V-shape. The section at Filchner Trough (Fig. 3b), a dense-shelf region, shows a strongly pronounced V-shape in which the 27.7 kg m$^{-3}$ isopycnal is found approx. 500 m deeper above the continental slope than on the continental shelf. The southern arm of the V-shape is pronounced in the Filchner Trough, as this is an area of DSW overflow (Fig. S1). Further to the west at the Ronne Trough (Fig. 3c), water of colder temperature reaches deeper than at the Filchner Trough. However, the V-shape in the isopycnals is less pronounced in the upper open ocean, i.e., the region displays a less pronounced northern arm, due to the deepening of the warm core of the mWDW transported by the ASC along its path along the continental slope (Fig. 3). At the Antarctic Peninsula in the western Weddell Sea (Fig. 3d), a V-shape in the isolines is clearly visible only in the temperature field and for the 27.7 kg m$^{-3}$

isopycnal and higher-density but not for lower-density isopycnals. In addition, mWDW at temperatures around -1°C can be found on the continental Shelf and in the Larsen Ice Shelf cavity (Fig. 3c, d). This water mass originates from Ronne Trough and follows the coast northward (Fig. S1a). The Filchner Trough has previously been shown to be an entrance point for mWDW to reach the Filchner Ronne ice shelf (e.g. Foldvik et al., 1985; Ryan et al., 2017). Because of its particular relevance, we will focus on the Filchner Trough in our further analysis. Generally, the circulation in REF on and off the continental shelf are

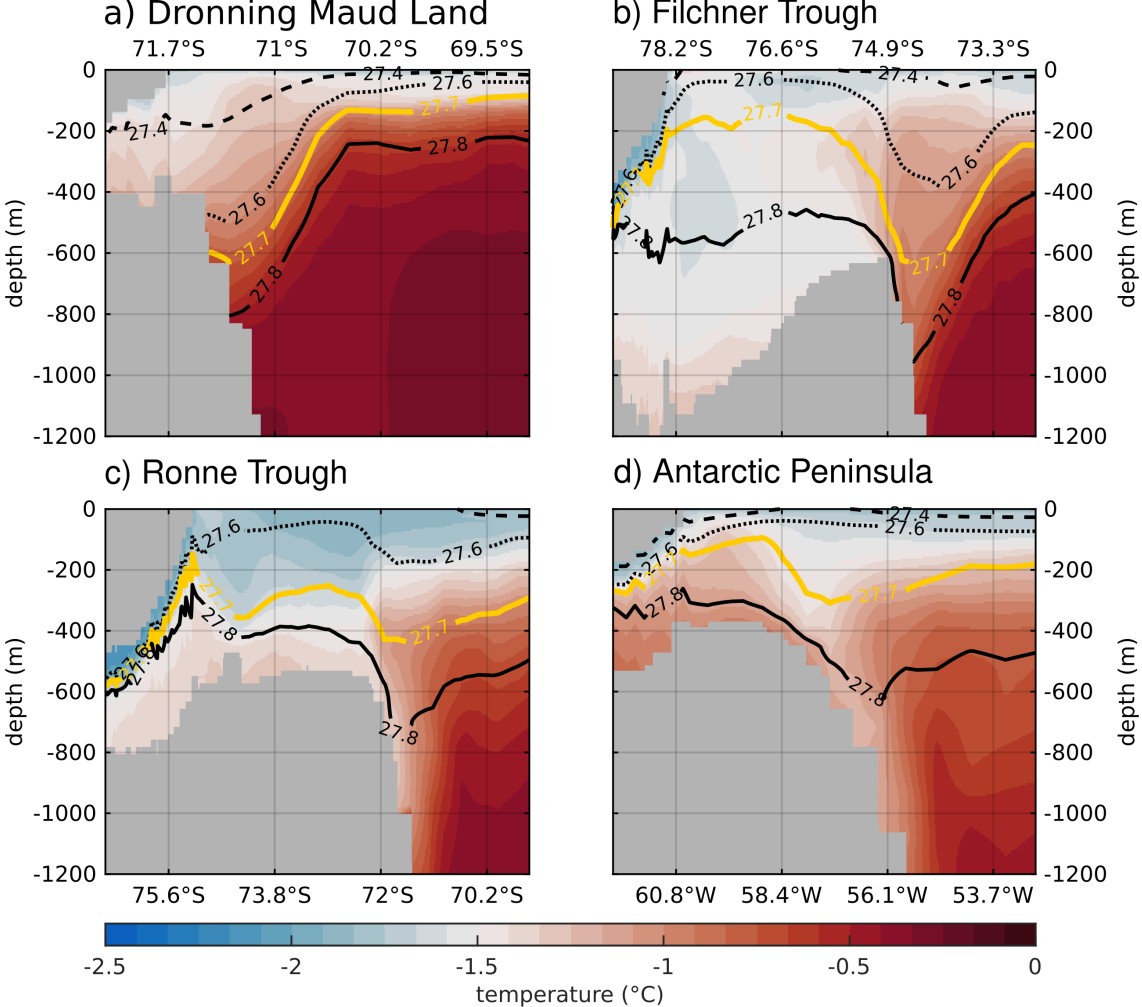

**Figure 3.** Potential temperature sections across the continental slope in the southern Weddell Sea with isopycnals shown relative to 1000 kg m$^{-3}$ for 27.4, 27.6 27.7 (orange) and 27.8 kg m$^{-3}$ for the average over 2010-2014 in REF. a) Dronning Maud Land, b) Filchner Trough, c) Ronne Trough, and d) Antarctic Peninsula. The continent and ice shelves (marked in grey) are on the left-hand side for all panels. See Figure 1 for the exact locations of the transects.

well represented by the model. Comparison to observations from mooring and CTD data (Schröder, 2010; Schröder et al.,

2014, 2016, 2019; Janout et al., 2019) show that the salinity on the continental shelf is slightly underestimated by the model results by up to 0.15 psu (Fig. S2b), however some areas in particular in Filchner Trough can overestimate the salinity slightly. The temperature on the shelf is up to 1°C warmer in the model than in observations (Fig. S2a), but largest differences in the temperature can be found at the eastern slope of the Filchner Trough where the highly variable mWDW current flows onto the continental shelf. At the continental slope, the temperature difference can be larger, while the mWDW transported by the

ASC is colder by up to 0.5°C. FECO produces a similar temperature distribution as REF (Fig. S3a), but produces lower salinity values (Fig. S3b).

## 3.2 Present-day seasonality of the V-shape at Filchner Trough (REF simulation)

Previous studies have shown that seasonal variations in the density distribution at the continental shelf break control the on-shore flow of mWDW (e.g. Teske et al., 2024). During the historical 2010-2014 time period in REF, the depth and steepness

of the V-shaped isopycnals show seasonal variations (Fig. 4). More intense sea-ice formation on the continental shelf leads to thicker sea-ice above the continental shelf (Fig. S4) and stronger DSW export in winter (June-August) and spring (September-November) than in summer (December-February) and autumn (March-May), which pushes down the isopycnals at the continental slope by approx. 50 m (Fig. 4 a, b). Such a vertical distance corresponds to two to three layers in the ocean grid. The seasonality is consistent with observed variations in the thermocline depth at the Filchner Trough sill, though the variation is

low compared to the amplitude of over 100 m observed (Hattermann, 2018). The isopycnals reach the shallowest position in autumn. The minimum sea ice extent is reached in general in February. While near-coastal sea ice formation starts again in March, the increased export of DSW from the continental shelf is delayed due to the distances involved between the area of sea ice formation along the coasts and the continental slope (see also Fig. S8e-h). The isopycnals at the continental slope therefore only deepen later, leading to the shallowest maximum depth being reached in autumn. Sea-ice melting during summer reduces

the steepness of the arm of the V-shape above the continental shelf (Fig. 4c). Subsequently, stronger densification over the continental shelf through sea-ice formation in autumn steepens the onshore arm again (Fig. 4d). Sea-ice formation in winter is particularly pronounced along the coasts of the Weddell Sea (Fig. S5e), while in summer most of the Weddell Sea is dominated by sea-ice melting (Fig. S5i). Only a small band along the Filchner-Ronne Ice Shelf edge produces sea ice year round, which - together with sea-ice deformation by tides and variable winds - leads to a band of exceptionally thick sea ice north of Ronne

Ice Shelf in summer (Fig. S4i).

The rise of the thermocline in summer also enables seasonal pulses of mWDW to flow onto the continental shelf (see also Ryan et al., 2020; Teske et al., 2024). The meridional temperature gradient at 300 m depth is strongest during winter (Fig. S6) - which is consistent with observations (Pauthenet et al., 2021) - however the northern arm of the V-shape has its steepest angle during summer (blue line in Fig. S7). During summer, reduced DSW export (Fig. S8) and weakened Ekman downwelling (see

also Fig. 6: positive surface stress curl corresponds to Ekman downwelling on the southern hemisphere.) reduce the depth of the V-shape compared to winter (Fig. 4c). The main mechanisms behind the seasonal variations are thus the seasonal production of DSW and seasonally varying Ekman downwelling. These mechanisms as found in the simulation agree with observations and previous modelling studies (Peña-Molino et al., 2016; Le Paih et al., 2020; Pauthenet et al., 2021; Ryan et al., 2020).

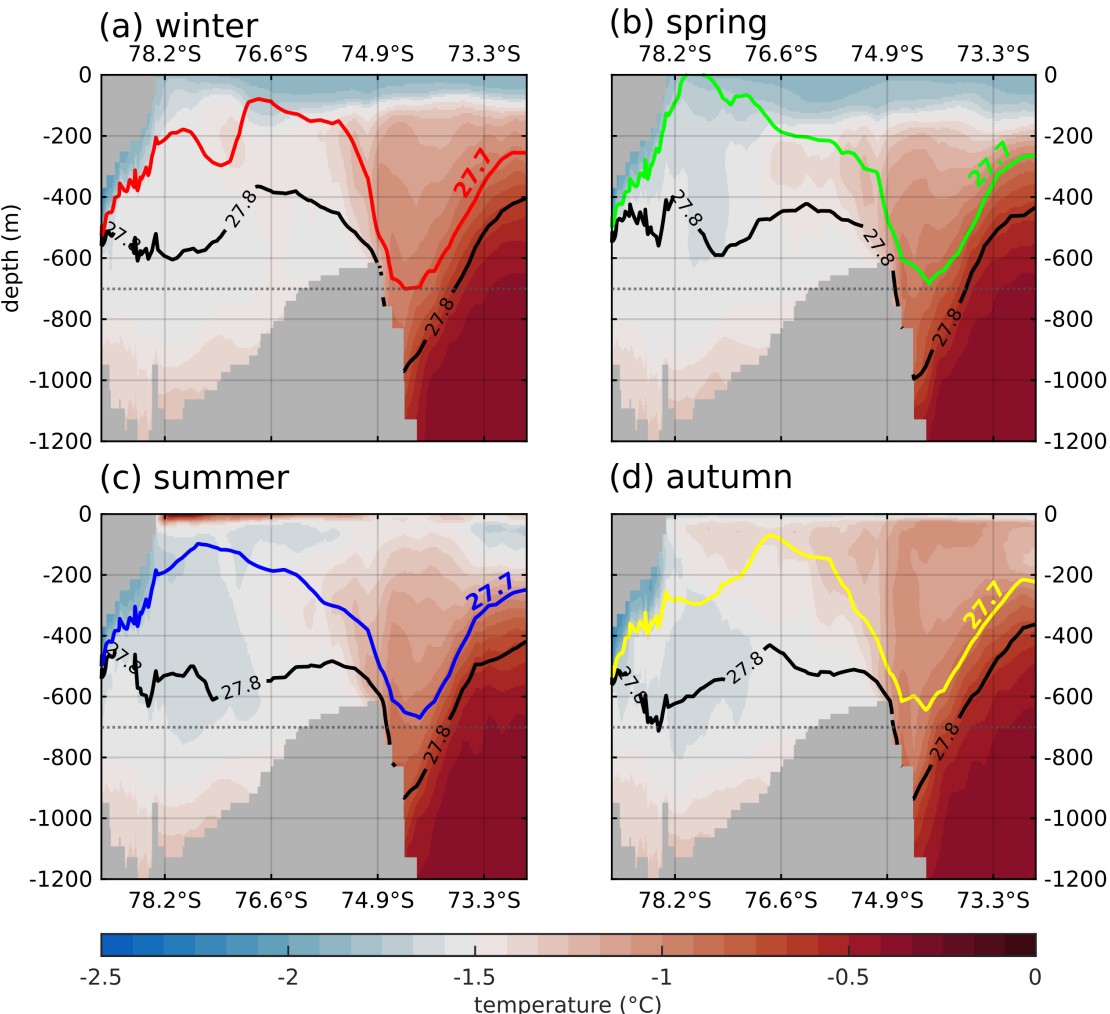

**Figure 4.** Potential temperature in the 2010-2014 mean of the Filchner Trough section in REF for winter (June-August), spring (September-November), summer (December-February), and autumn (March-May). The coloured (black) lines show the position of the $27.7 \, \text{kg m}^{-3}$ ($27.8 \, \text{kg m}^{-3}$) isopycnal in the respective season. Horizontal dotted line shows the maximum depth of the $27.7 \, \text{kg m}^{-3}$ isopycnal in winter. The continent and ice shelves (marked in grey) are on the left-hand side for all panels. See Figure 1 for the exact locations of the transects.

## 3.3 Effect of a warming climate on the seasonally varying V-shape geometry (REF simulation)

Over the 21st century, increased summer sea-ice melting and reduced freezing rates in winter above the continental shelf (Fig. S5) lead to freshening of the continental shelf and to a density redistribution in the Filchner Trough and across the continental slope. While the geometry of the V-shaped density distribution is mostly symmetrical in 2010-2014 (Fig. 4), it becomes asymmetrical and shallower towards the end of the century in the REF simulation (Fig. 5). A reduction of the density over large parts of the water column in Filchner Trough by up to $0.2 \, \text{kg m}^{-3}$ between 2000 and 2100 reduces the vertical extent

of the southern arm of the V-shape. In addition, a shoaling of the ASC over the course of the century Fig. S9) reduces the depth of the V-shape so that the yearly mean depth of the 27.8 kg m$^{-3}$ isopycnal lifts from approx. 992 m to 791 m between 2010-2014 and 2096-2100 (dotted and dashed horizontal lines in Fig. 5, respectively). Of all seasons, the V-shape reaches its deepest position in summer by the end of the century (Fig. 5c), as opposed to in winter for the 2010-2014 period (Fig. 4).

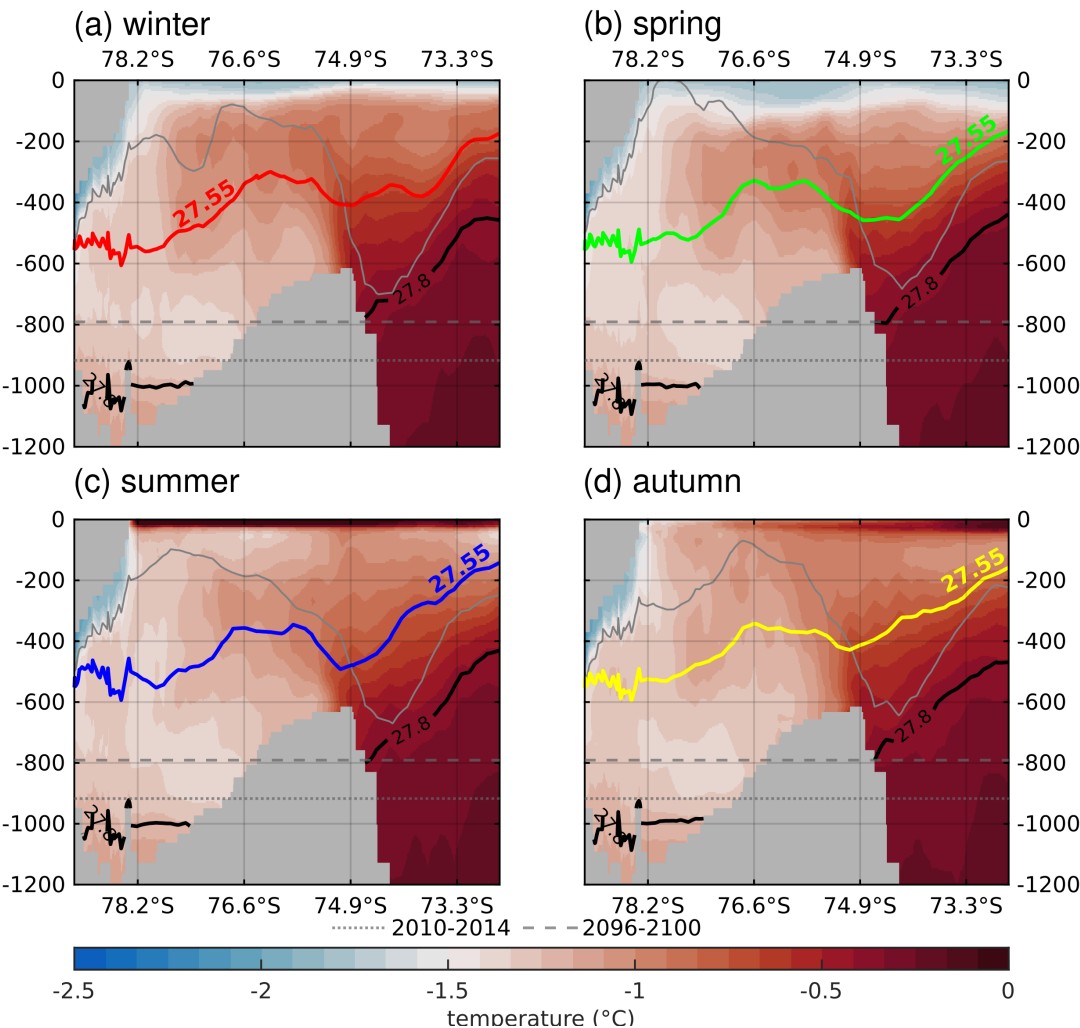

**Figure 5.** Potential temperature in the 2096-2100 mean of the Filchner Trough section in REF for a) winter (June-August), b) spring (September-November), c) summer (December-February), and d) autumn (March-May). The coloured lines show the position of the 27.55 kg m$^{-3}$ isopycnal in the respective season, the black line the 27.8 kg m$^{-3}$ isopycnal. The gray line shows the position of the 27.7 kg m$^{-3}$ from Fig. 4. The dotted (dashed) horizontal gray line shows the mean depth of the 27.8 kg m$^{-3}$ isopycnal in the multi-year mean of 2010-2014 (2096-2100).

The change in sea-ice thickness and concentration (Fig. S10a & b) during the 21st century does not only lead to freshening of shelf waters, but also influences how the wind field impacts the ocean surface. To isolate the relative effects of wind speed changes and changes in sea-ice properties, we compute the wind stress curl from the 2m wind field assuming a constant drag coefficient over ocean ($C_{AO}$=1.00x10$^{-3}$) and sea ice ($C_{AI}$=1.32x10$^{-3}$), weighted by sea ice concentration per grid cell. We find that an intensification of the wind field towards the end of the century enhances existing wind stress curl patterns in winter (Fig. 6a-d), while qualitatively the patterns remain largely unchanged. In contrast, considering the ocean surface stress, i.e. taking the effect of the sea-ice cover into account, we find an alternating pattern of negative and positive surface stress curl, corresponding to alternating areas of up- and downwelling, respectively (Fig. 6e-h). With a reduction of winter sea-ice thickness by at least 35% - in large areas even more - compared to the beginning of the century, some of the up- and downwelling areas are redistributed. Variability along the profile is slightly reduced so that the surface stress curl resembles the wind stress curl more closely at the end of the century (compare Fig. 6c/g). Similar changes in the wind stress curl can be seen in spring and autumn (Fig. 6 b/d vs. f/h). In summer, when the sea-ice thickness (Fig. S5) and concentration (not shown) in the Filchner Trough is already low between 2010 and 2014, a northward shift of the wind field increases areas of downwelling above the continental slope (Fig. S11). The impact of sea ice on the surface stress curl is especially visible in autumn when the Filchner Trough is partly covered by sea ice with northward decreasing thickness (Fig. S11d). South of approx. 76°S, sea ice creates alternating patterns of positive and negative surface stress curl, while north of this, surface stress curl closely resembles the wind stress curl (Fig. 6 d, h).

From the fact that in the comparison between 2096-2100 and 2010-2014 we find modified patterns of spatial variability in the ocean surface stress curl (lower panels in Fig. 6) much more pronounced than in the wind stress curl (upper panels in Fig. 6), we conclude that the up- and downwelling patterns are created by the sea ice distribution rather than the wind field. The combination of reduced DSW export in winter and spring and the enhanced wind stress impact on the ocean is then responsible for the change in the seasonal variations of the V-shape.

### 3.4 Sensitivity of the Filchner Trough circulation to atmospheric forcing

The application of the regional higher-resolved CCLM forcing in FECO affects the heat transport onto the continental shelf and leads to a regime shift in the Filchner Trough circulation from a DSW dominated to a mWDW dominated state in 2093, as seen by the much warmer shelf temperatures at the end of the century (Fig. 7c). Reduced freezing rates along the coasts in winter and higher melt rates in summer (Fig. S5) due to higher air temperatures (not shown) compared to REF lead to a decrease of the mean salinity in the Filchner Trough by up to 0.2 psu over the course of 10 years. During this timespan, all three time slices converge to a new mean state (Fig. 7d). Additionally, lower wind speeds over most of the western Weddell Sea in FECO reduce the wind stress on sea ice and ocean, except for some areas at the eastern Ronne Ice Shelf front and the Filchner Ice Shelf. Here, stronger off-shore winds dominate (Fig. S12). In contrast to the salinity, the mean potential temperature does not appear to be sensitive to the forcing change and no obvious change in temperature was found during the 10-year transition times in FECO (Fig. 7c). In 2093, mWDW enters the Filchner Trough as a near-bottom current across the sill (Fig. 8d), strongly increasing the temperature in the trough.

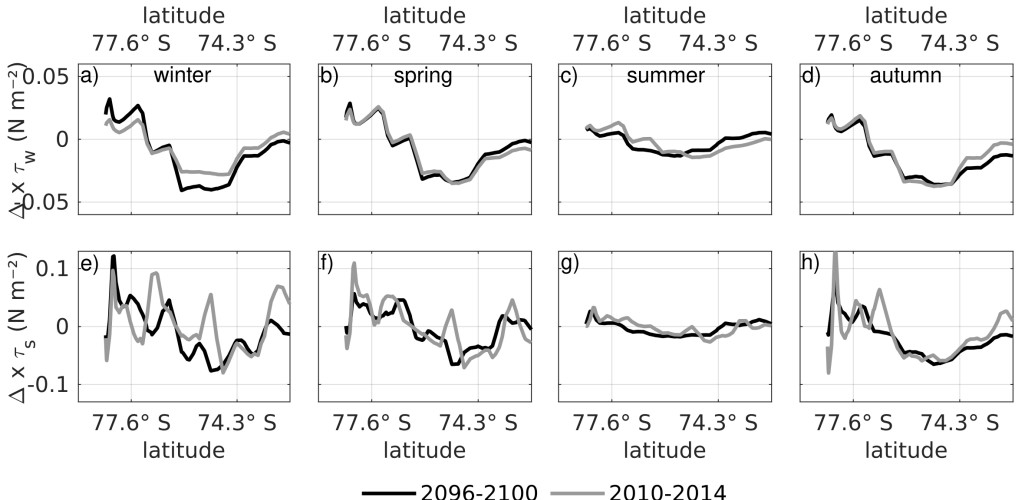

**Figure 6.** Wind stress curl ($N\,m^{-3}$), estimated from the 2 m-wind field, in the 5-year means of 2096-2100 (black line) and 2010-2014 (gray line) in REF above the Filcher Trough section (see Fig. 1) for (a) winter (June-August), (b) spring (September-November), (c) summer (December-February), and (d) autumn (March-May). (e)-(h) Same as (a)-(d) but for surface stress curl on the liquid ocean from overlying atmosphere and sea ice. Positive curl corresponds to downwelling.

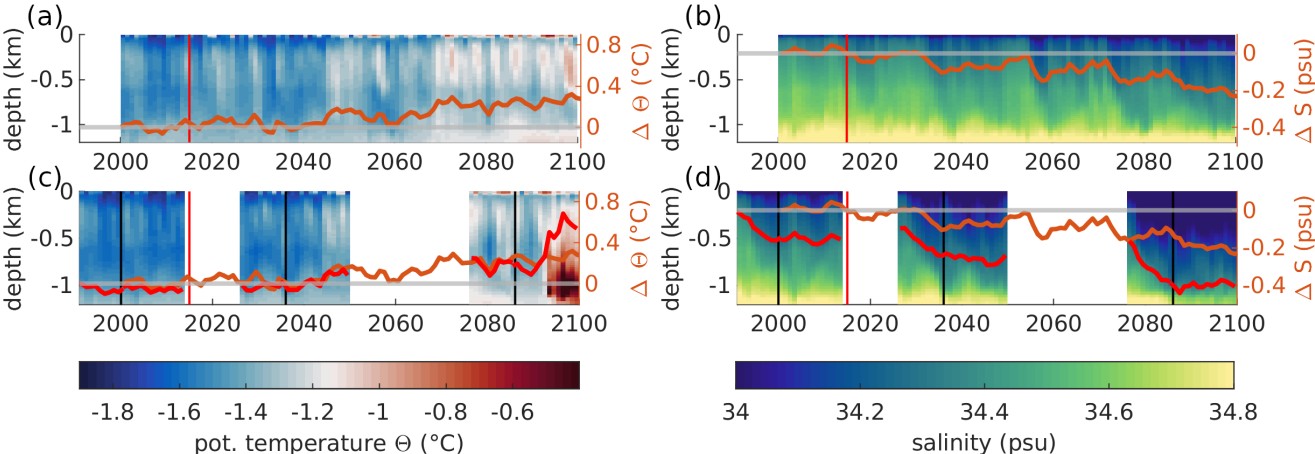

**Figure 7.** Hovmöller diagram of the yearly mean temperature in Filchner Trough (see Fig. 1) for (a) REF and (c) FECO. Coloured lines show the relative temperature change of the horizontally and vertically averaged temperature of the whole water column compared to 2000. Orange lines are for REF and red lines for FECO.(b) & (d) Same as (a) & (c) but for salinity. Red vertical line: boundary between historical and future climate scenario. Black vertical lines: boundary between transition time and experiment in FECO.

Previous studies have suggested that warmer and longer-lasting pulses of mWDW entering the Filchner Trough can be a precursor for a regime shift in the Filchner Trough from a DSW-dominated to a mWDW-dominated state (Timmermann and Hellmer, 2013; Timmermann and Goeller, 2017; Ryan et al., 2020; Nissen et al., 2023; Teske et al., 2024), but despite an increase in the mean temperature and visible mWDW pulses in the trough, no sudden warming of the Filchner Trough occurs in REF. Over the course of the 21st century, REF shows increasingly stronger seasonal pulses of mWDW crossing into the Filchner Trough across the shelf east of Filchner Trough and along the eastern slope of the trough (Fig. S13). The temperature of mWDW carried by the ASC shows a warming trend of approx. 0.053°C per decade between 2000 and 2100. This agrees well with the 0.05°C per decades that were observed between 1980 and 2010 (Schmidtko et al., 2014). From our results it is unclear, if the warming trend is a result of a warming of the source of the mWDW in the ACC or a result of reduced modification of the WDW while being transported into the southern Weddell Sea resulting in warmer mWDW reaching the Filchner Trough sill.

Before the regime shift occurs in FECO, the reduced density of the upper 1000 m in the open ocean leads to changes in the position and symmetry of the V-shaped isopycnals above the continental slope (Fig. 8). The comparison of the seasonal positions of the deepest point of the 27.4 kg m$^{-3}$ isopycnal in FECO for 2088-2092 (Fig. 8b) - chosen for its position at depth above the continental slope at all times, but nearly no intersection with the surface in REF - yields an additional horizontal component to the seasonal depth variation seen in REF (Fig. 8a). This horizontal movement displaces the V-shape by approx. 50 km between summer, when the deepest point of the V-shape is at its southernmost point, and winter. The deepest point of the V-shape is closest to the sill in summer and furthest away in winter (Fig. 8b). In addition, the V-shape reaches its deepest position in spring and summer, while REF reaches the deepest point in winter and spring. The fresher water masses on the continental shelf in FECO cause the southern arm of the V-shape to be very flat and to vanish during autumn (Fig. 8b, yellow curve). Stronger Ekman downwelling in autumn, but a late onset of the freezing season lead to a situation that temporarily resembles fresh-shelf regions like the Dronning Maud Land section (Fig. 3a) where the V-shape is missing its on-shore arm and the isopycnals cancel at the slope. Sea-ice formation in winter regularly restores the V-shape.

### 3.5 Accelerated density redistribution with high-resolution atmospheric forcing

The change to a high-resolution atmospheric forcing does not only affect the slope front symmetry and seasonality, but also accelerates the reduction of density in the Filchner Trough over the 21st century compared to REF (Fig. 9). The differences in the density evolution between REF and FECO are particularly large in the Filchner Trough (Fig 9a). Here, the application of CCLM forcing leads to a negative density trend between 2000 and 2092 that is 1.47 times larger than in REF. Values after 2092 were excluded from the calculation due to the impact the regime shift has on the density of the Filchner Trough. In contrast, above the continental slope north of the Filchner Trough sill, the trend is slightly smaller in FECO than in REF (-1.5 kg m$^{-3}$ century$^{-1}$ as compared to -1.6 kg m$^{-3}$ century$^{-1}$; Fig. 9b). Separately assessing the trends in temperature and salinity reveals that the density trend in the trough is driven by freshening due to reduced freezing rates along the coasts in winter and higher melt rates in summer, while the trend in the mWDW transported by the ASC is driven by a combination of warming and freshening (Fig. S14). The change in density is larger in FECO in the Filchner Trough than at the off-shore

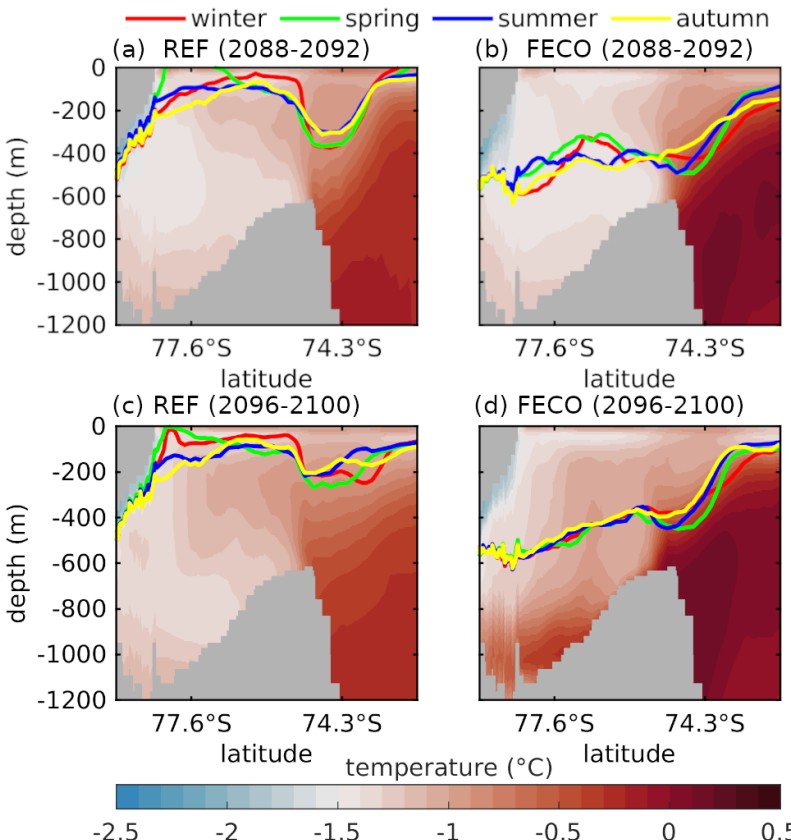

**Figure 8.** Potential temperature in the 2088-2092 mean (5 years before regime shift in FECO) in a) REF and b) FECO. Coloured lines show the position of the 27.4 kg m$^{-3}$ isopycnal. c) and d) same as a) and b) but for 2096-2100.

location. From this, we conclude that reduced sea-ice formation is not the only factor influencing the density distribution across the continental slope, but the dominating one for the existence of V-shaped cross-slope isopycnals.

### 3.6 Influence of cross-slope currents on the V-shaped density distribution

With the onset of the near-bottom current of mWDW entering the Filchner Trough in FECO in 2093, the V-shaped density distribution at the continental slope experiences pronounced structural changes that lead to the near-loss of the southern arm of the V-shape. Visible as a sudden increase in the average temperature in the Filchner Trough (Fig. 7c), and as a layer of warm water at the bottom of the Filchner Trough (Fig. 8d), the inflow of mWDW in FECO in 2093 brings the bottom density in the Filchner Trough closer to that of the ASC. This has the effect of increasing the stratification of the water column. In combination with low sea-ice formation rates and reduced mixing (not shown) during the freezing season, the seasonal variations in the southern arm of the V-shape vanish at depths below approx. 450 m (Fig. 8d). In REF, the density distribution on the continental shelf leads to the formation of the typical V-shape(Fig. 8c). The isopycnals above the continental shelf

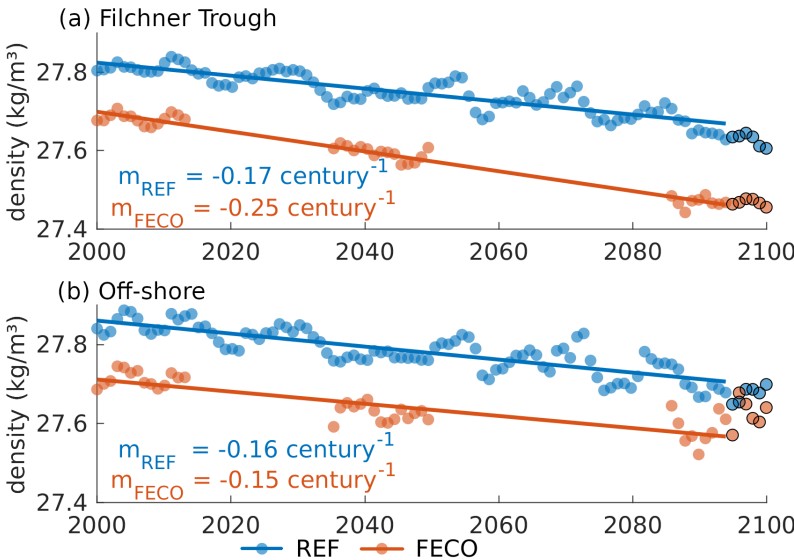

**Figure 9.** Linear regression of the annual mean potential density relative to 1000 kg m$^{-3}$ over time in a) the Filchner Trough and b) at the Filchner Trough Sill at 636 m depth for REF in blue and FECO in red. The slope of each linear fit is given in the figure. Data points outlined in black were excluded from the calculation due to the regime shift in 2093. The areas over which properties are averaged for the time series are indicated in Fig. 1.

also experience stronger seasonal depth variations due to the location closer or in the mixed layer. Seasonal variations of the depth and position of the V-shape and the position of the northern arm can be found in FECO and REF. The on-shore arm of the V-shape in FECO decreases strongly in vertical extent. From a height difference in spring of approx. 200 m between the
310 deepest point of the V-shape and the shallowest point above the continental shelf, the 27.4 kg m$^{-3}$ isopycnal position reduces its vertical extend to a range of approx. 80 m after the bottom current onset. With the near-bottom inflow of mWDW into the trough, the sensitivity of the southern arm of the V-shape to sea ice formation is reduced, while the northern arm, which is created by the wind, remains. The weaker stratification in REF has the effect of making the on-shore arm shallower than 450 m more pronounced. Below this depth, the isopycnals take on a similar shape as in FECO.
The timing of the intensified mWDW flux in FECO can be related directly to the density gradient across the Filchner Sill and the depth of the thermocline at the sill. In the following, the thermocline is defined as the greatest vertical temperature gradient. In FECO, the thermocline remains above the Filchner Trough sill depth throughout the FECO simulation from 2086 to 2100 and stays between 400 m to 600 m depths, while the thermocline in REF rises from approx. 1000 m in 2000 (Fig. S9) to above 600 m depth over the course of the 21st century (last 15 years shown in Fig. 10a). The sill depth of approx. 600 m
(horizontal gray line in Fig. 10a) is only crossed occasionally and never for an extended period of time except for the last four years. Despite the thermocline in FECO already reaching above the sill from the beginning of the time slice 2086-2100, the warm near-bottom current is still only developing at a later date. This agrees with the conclusions presented by Haid et al.

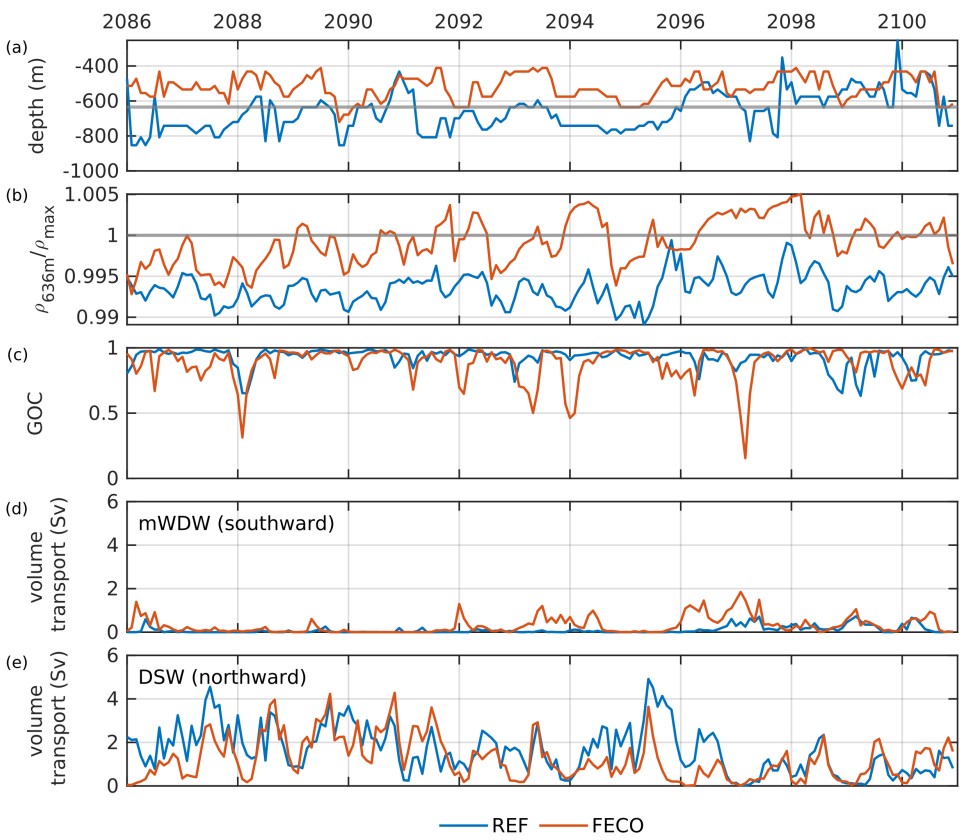

**Figure 10.** a) Depth of the thermocline at the Filchner Sill (horizontal line: sill depth at 636 m) and b) density ratio between the maximum density in Filchner Trough ($\rho_{max}$) and the mean density at the Filchner Trough sill (see Fig. 1) at 636 m depth ($\rho_{636m}$). c) Time series of the grade of connectivity (GOC) of the V-shaped density structure above the continental slope at the Filchner Trough sill (for area outline see Fig. 1; for calculation see section 2.3). d) Southward volume transport in Sv across Filchner Sill Central (see Fig. 1) of water masses mWDW with a potential temperature of $\Theta > -0.8$°C and e) northward volume transport of DSW ($\Theta < -0.8$°C). REF is presented in blue, FECO in orange.

(2023) suggesting that another key factor for or against an inflow of mWDW onto the continental shelf in addition to the depth of the mWDW at the slope is the density ratio between the dense water on the shelf and the mWDW of the ASC (see also Nissen et al., 2023). FECO only experiences the regime shift after the density ratio between the DSW of the Filchner Trough and the mWDW in the ASC changes in favour of the mWDW (Fig. 10b). And while the thermocline in REF does also cross sill depth during the last five years of the simulation, the density ratio remains in favour of the DSW, therefore preventing the intrusion of large amounts of mWDW into Filchner Trough.

The inflow of mWDW onto the continental shelf is not only visible in the cross-slope density structure, but also as a disruption of the V-shape in along-slope direction. In both simulations, the GOC at the Filchner Sill shows multiple minima between 2086 and 2100, demonstrating phases of instability of the Antarctic Slope Front where the barrier across the sill is weakened (Fig. 10c). However, not all of the recognised events of a minimum GOC are related to an increased flow of mWDW into the Filchner Trough. After 2093, the southward mWDW ($\Theta$ >-0.8°C) transport (Fig. 10d)) and the GOC are significantly correlated with a 3 month lag in REF (p=7x10$^{-7}$, r=0.36) and in FECO (p=0.0005, r=0.35). There is also a weak correlation between the outflowing DSW ($\Theta$ <-0.8°C; Fig. 10d) and the GOC in REF (p=0.0001, r=0.29) while such correlation is missing in FECO (p=0.27, r=0.08). The changing circulation patterns associated with the density reorganization on the shelf in FECO remove the weak correlation between the water export and GOC that can be found in REF. The calculation of the GOC was performed for 250 m depth, where the V-shape is known to be present and the layer does not intersect with the continental slope at the approximate location of the V-shape. Below the mixed layer, the GOC is robust against the depth selection within a certain depth range spanning between approx. 100 m near the surface to approx. 200 m below the depth of the Filchner Trough sill.

## 4    Discussion

### 4.1    Drivers of the accelerated density redistribution: wind speed and sea ice

Our results show that for the high-emission scenario SSP3-7.0, lower sea-ice production in a simulation with high-resolution atmospheric forcing data accelerate the density redistribution on the continental shelf of the southern Weddell Sea relative to a simulation with lower-resolution atmospheric forcing. Subsequently, the accelerated density redistribution induces a regime shift in the Filchner Trough area before the end of the 21st century. The regime shift from a DSW-dominated to a mWDW-dominated state is accompanied by the near-loss of the southern arm of the V-shaped density distribution at the continental slope that prominently separates the shelf from the open ocean in today's climate.

The main factors controlling the circulation regime in Filchner Trough are the wind field and sea-ice production (Teske et al., 2024), which directly influence the density distribution on the shelf. The on-shore transport of mWDW, which intensifies in the case of a regime shift, strongly depends on the density ratio across the continental slope at the Filchner Trough sill. This ratio changes faster in FECO than in REF, driven by an accelerated freshening of the continental shelf in response to higher summer temperatures and weaker wind speeds in CCLM. Higher summer air temperatures in FECO compared to REF lead to a stronger decline and local melting of summer sea-ice. Since multi-year mean air temperatures are actually lower in FECO than in REF, this emphasizes the importance of the seasonal cycle.

The dynamic downscaling of the wind field in the CCLM simulation produces wind differing from those in the AWI-CM fields. In particular, the decrease in wind speed between CCLM and AWI-CM can be attributed to the better resolved topography of the Antarctic Peninsula. The lower resolution of AWI-CM leads to smoothing of the topography of the Antarctic Peninsula, which results in mountains not being well resolved and in westerly winds being too strong in the AWI-CM forcing data (Stössel et al., 2011). Weaker wind speeds in FECO may have two effects: (i) smaller heat loss to the atmosphere, and (ii) less wind-driven sea-ice transport. Both effects tend to reduce sea-ice production and thus winter brine release in regions

that are dominated by freezing and export of sea ice. In contrast, stronger off-shore winds at the Filchner Ice Shelf front in FECO increase the frequency and size of coastal polynyas that locally cause higher sea-ice production rates. While this could counteract some of the freshening of the Filchner Trough, the net effect appears to be dominated by the weaker off-shore winds in the high-resolution atmosphere model (CCLM).

In the historical time period between 2010 and 2014, REF produces a steeper, narrower temperature and density gradient across the ASF at 300 m depth during winter than during summer, which agrees with the assessment of the ASF by Pauthenet et al. (2021). According to their study, the ASF forming the northern arm of the V-shape has the smallest meridional extent during winter months, which translates to a strong temperature gradient. The steepening is a result of stronger DSW export in winter. The entrainment of colder, fresher surface water suppresses the isopycnal and forces the isopycnal and thermocline into a steeper position. However, observational data for winter in the ice-covered southern Weddell Sea are scarce, leading to high uncertainties in the assessment of the position and extent of the ASF in the southern Weddell Sea. Additionally, Pauthenet et al. (2021) applied their analysis at 300 m at relatively shallow depths for the Weddell Sea, where the pycnocline extends deeper than 380 m (Thompson et al., 2018). As Fig. 4 has shown, the isopycnals are steeper in greater depth. It is therefore unclear if the shift in the minimum extent of 1.53° of the distance between the -0.7°C and -0.3°C isotherms at 300 m depth in winter during 2010-2014 of REF to a minimum extent of 5.41° in summer in 2096-2100 (Fig. S6b) is an effect of climate change and the shift in the front dynamics.

The decreasing wind strength over the 21st century in REF reduces the Ekman transport and downwelling above the continental shelf. Multiple studies have shown that a strengthening and poleward shift of the westerlies and a weakening of the easterly winds above the continental shelf will weaken the Ekman downwelling along the coast of the Weddell Sea (Spence et al., 2017; Naughten et al., 2021; Teske et al., 2024). The indirect influence of the wind strength on the flow of Antarctic Surface Water and mWDW onto the continental shelf has also been shown in previous studies (Hattermann et al., 2014; Dinniman et al., 2015; Haid et al., 2015; Hatterman, 2018). In particular, Hattermann et al. (2014) described the weakening of shallow and the strengthening of deep inflow towards the Fimbul ice shelf cavity in response to weaker easterly winds based on observations and modelling results. The projected reduced wind speed of the easterlies in our model towards the end of the 21st century and the southward shift of the mid-latitude westerlies, consistent with Spence et al. (2014) and Goyal et al. (2021), broaden the V-shaped density structure at the continental slope and reduce its depth in REF, as seen in Fig. 3. The uplift of the thermocline due to weakened downwelling lifts mWDW closer to the sill depth and slowly increases the amount of mWDW flowing into the Filchner Trough. A similar signature is found when comparing results from FECO with those from REF: The generally weaker wind speeds in the southern Weddell Sea in FECO than in REF weaken the Ekman downwelling further, reducing the V-shape depth in comparison to REF. The reduced density gradient weakens the barrier effect of the cross-slope gradient further and allows the strong cross-slope current to develop towards the end of the century that does not develop in REF.

The warming of the Filchner Trough through increased heat transport creates a warm shelf that, in some parts, resembles the warm shelf in the Amundsen Sea (Thompson et al., 2018). The exception is the continued presence of the V-shape in the density distribution of the upper ocean at Filchner Trough. Atmospheric patterns like the Amundsen Sea Low create a more

variable surface wind (Turner et al., 2013), and the absence of DSW formation on the shelves of the Amundsen Sea remove the mechanisms for the formation of the V-shaped density distribution.

## 4.2 Drivers of the accelerated density redistribution: cross-slope density gradient

One of the main criteria for the potential of the current system in the Filchner Trough to switch from a DSW-dominated state to the mWDW-dominated state is the density gradient across the Filchner Trough sill as has been previously shown (Ryan et al., 2017; Haid et al., 2023; Nissen et al., 2023). A near-bottom current of mWDW onto the continental shelf only develops once this gradient flattens out or reverses. Our results show that this on-shelf current disrupts the V-structure of the isopycnals at the slope, reducing the V-shape to the upper 450 m of the water column and warming the Filchner Trough. The horizontal distance

of the ASF (northern arm of the V-shape) to the continental slope varies over the year as shown in an analysis of observational data by Pauthenet et al. (2021).

The introduction of the GOC provides a metric for the stability of the ASF in areas of cross-slope transport in dense-shelf areas. However, not all features of the V-shape of the density distribution are included in its definition. Because it searches for density minima at a chosen depths, inferring the existence of the complete V-shaped distribution is not possible from the GOC

alone. This information needs to be compiled before applying the algorithm. The GOC is useful to detect strong changes in the cross-slope density distribution that remove the dip in the isopycnals completely or lead to a lateral displacement of the V-shape in a section parallel to the continental slope. However, the GOC is not able to differentiate between a lot of small displacements that just exceed the distance threshold $d_L$ and few, large displacements. The selection of $d_L$ as a maximum displacement threshold also influences the result. A distance larger than ten times the grid size reduces the number of recognised events by

12.5% compared to a $d_L$ of two times the grid size; with $d_L$ 100 times the grid size, the number of recognised events decreases by 29.1%. Additionally, the GOC is sensitive to the chosen depth. Generally, the GOC is robust against depth selection within the water column below the mixed layer but above the depth of the continental shelf to prevent an intersection of the isopycnals with the sea floor within the V-shape. To minimize the impact of seasonal variability in the upper ocean and to avoid an intersection of the chosen depth layer with the continental slope within the main feature of the V-shape, a depth of 250 m was

chosen for this study.

The three-month delay between intensified cross-slope currents and a minimum event of the GOC at 250 m depth indicates a connection based on seasonal cross-slope water transport. Ryan et al. (2017) showed that during winter intensified DSW production and export subduct the isopycnals at the continental slope, reducing or stopping on-shore transport of mWDW. This conclusion is back up by the lack of correlation in FECO where DSW production is reduced compared to REF. The lack

of non-delayed correlation between cross-slope currents and the GOC indicates that an intensified bottom current does not destabilize the ASF at depths shallower than the sill. This also implies that a (hypothetical) decreasing trend in the GOC cannot be regarded as an indicator that a cold-to-warm regime shift in Filchner Trough is about to occur in the near future.

### 4.3 Comparison to other studies

Considering our results in the context of other scenario simulations for the 21st century, it becomes clear that the choice of
atmospheric forcing data has a strong impact on the susceptibility of Filchner Trough to undergo a regime shift before 2100.
Previous future projections produced a tipping of the Filchner Trough circulation in very high-emission scenarios (e.g., SSP5-
8.5 or similar; Timmermann and Hellmer, 2013; Nissen et al., 2023) or idealized scenarios (e.g., 1pctCO2 scenario; Naughten
et al., 2021). Ocean simulations with lower emission scenarios generally did not produce a regime shift (Timmermann and
Hellmer, 2013; Nissen et al., 2023), except in case of a coupled atmospheric component (Teske et al., 2024, regime shift
in SSP3-7.0 and SSP5-8.5 scenarios). While our reference simulation for the emission scenario SSP3-7.0 does not produce a
regime shift during the 21st century, downscaling of the atmosphere in FECO induces a strong mWDW inflow before the end of
the 21st century. We therefore conclude that the potential for a regime shift in a warming climate in the SSP3-7.0 scenario might
be higher than suggested by previous studies using relatively low-resolution atmospheric forcing (Nissen et al., 2023). With the
currently implemented policies, global warming is projected to be between 2.2°C and 3.5°C by 2100, which corresponds to the
warming projected for the SSP2-4.5 (2.1°C-3.5°C) and SSP3-7.0 (2.8°C-4.6°C) scenarios (Calvin et al., 2023). Our results thus
highlight the importance of developing a better understanding of the potential implications for vulnerable climate components
like the Antarctic Slope Front and the Filchner Trough in the current climate development.

### 4.4 Limitations and caveats

Based on the results in REF and FECO, the two atmospheric data sets impact the ocean in different ways. However, it is not
straightforward to cleanly separate the effects of a higher atmospheric resolution and the different atmospheric states the two
atmospheric models provide. It remains unclear whether the higher resolved model output from CCLM impacts the ocean
differently than the AWI-CM data because it simulates a different mean climate (e.g. warmer air temperatures) or because it
resolves small-scale variability in the atmosphere. We note that the differences in the wind field between CCLM and AWI-CM
can be explained by the improved representation of gradients in the orography of Antarctica (Mathiot et al., 2011; Elvidge et al.,
2014; Cape et al., 2015), supporting, at least in part, the importance of resolving small-scale atmospheric variability. Several
studies have suggested that eddies play an important role in the transport of Circumpolar Deep Water across the continental
shelf break (Nøst et al., 2011; Stewart and Thompson, 2015). Our model is not eddy-resolving, but it has an eddy-permitting
grid resolution on the Weddell Sea continental shelf (∼3-12 km). It has been proposed by Nøst et al. (2011) that freshening
of the shelf can lead to increased eddy kinetic energy which drives onshore cross-ASF eddy transport. The resolution of
approx. 12 km at the continental slope in our model grid might lead to an underrepresentation of eddies, the Rossby radius
in the southern Weddell Sea being 2-5 km. With reduced eddy presence, the model might underestimate cross-slope volume
transport. An underrepresentation of eddy-driven on-shore transport of mWDW would mean a greater depth of the V-shape
as the lack of mesoscale eddies reduces the relaxation of the isopycnals above the continental slope (Dettling et al., 2024).
As a result, the temperatures on the continental shelf would be underestimated. A shallower V-shape at the continental slope
in a simulation with an adequate eddy representation might lead to an earlier regime shift of the Filchner Trough due to the

lower density gradient into the shelf and the higher connectivity between the ASC and the Filchner Trough through the sloping isopycnals. Armitage et al. (2018) found that the southward contraction of Westerlies during positive SAM enhances northward Ekman transport, dropping coastal sea level and weakening the ASF. In negative SAM phases the opposite happens. While the impact of long-term positive SAM trends on the ASF are uncertain, modelling studies suggest that a drop in Antarctic coastal
sea level could weaken the ASF, leading to shoaling of the isopycnal on the continental slope (Spence et al., 2014, 2017). Similar events take place during El Niño events, where an anticyclonic atmospheric pressure anomaly over the Amundsen and Bellinghausen Seas lead to a weakening of the ASF in the Pacific Sector of the Southern Ocean (Armitage et al., 2018; Spence et al., 2014). ENSO should show their influence in REF and FECO in the same manner, because its origin in equatorial latitudes is not part of the CCLM domain and therefore any influence they have does not differ between REF and FECO. While
the analysis of remote connections is not included in this study, it should be considered in the future.

## 5 Conclusions

We have presented model results indicating the future evolution of the density distribution across the Filchner Trough sill, a key property of the Antarctic Slope Front in the Southern Weddell Sea. The reference simulation REF and the experiment simulation FECO with down-scaled atmospheric forcing both reproduce the typical V-shaped density distribution that is formed
by interplay of Ekman downwelling and Dense Shelf Water export in the southern Weddell Sea. Sea ice loss in the warming climate of the SSP3-7.0 emission scenario decreases the density on the continental shelf and flattens the southern arm of the V-shape in REF and FECO. This causes an increased sensitivity of the density distribution on the continental shelf to seasonal wind speed variations. Using results from a regional, high-resolution atmosphere model as forcing, we find an acceleration of the density redistribution on the continental shelf, which leads to a regime shift with modified Warm Deep Water entering
Filchner Trough before the end of the 21st century for the high emission scenario SSP3-7.0. Colder air temperatures in the multi-year mean are more than outweighed by the warmer summer air temperatures of the regionally down-scaled atmosphere, accelerating density loss on the continental shelf, which ultimately leads to a regime shift from a cold Dense Shelf Water dominated Filchner Trough to a warm, modified Warm Deep Water dominated trough in 2093. We could show that the application of forcing data with resolved mesoscale atmospheric processes increases the on-shore modified Warm Deep Water transport as
hypothesized. However, we observed a small decrease in the average Dense Shelf Water export from the shelf and no visible increase in the seasonality of the V-shaped density distribution above the continental slope. The criterion of spatial coherency of the V-shape along the continental slope, quantified by the grade of connectivity, is not usable as a tool to predict an imminent regime shift because the onshore transport of modified Warm Deep Water is, contrary to our expectations, not a result of a weakening of the slope front. Instead, the cross-slope transport leads to a temporary disturbance of the ASF and the associated
V-shape. While the density minimum is completely restored after disruption in present-day climate, in a warming climate the distinct V-shape remains confined to the upper ocean. Due to the geostrophic nature of the ASC, the flattening of the northern arm of the V-shape and stronger seasonality will affect the transport of the ASC and the transport of heat towards the peninsula and onto the continental shelf in the future.

The accelerated density redistribution by the high-resolution atmospheric data and the resulting regime shift show that the potential for a regime shift in the Filchner Trough in the SSP3-7.0 scenarios is higher than previously published ocean simulations of the same scenario suggest. With the current climate policies, the projected 21st-century global warming lies within the potential range of the SSP3-7.0 scenario, increasing the urgency to better understand and represent vulnerable climate components.

*Code and data availability.* The code for FESOM can be accessed from the esm-tool at https://github.com/esm-tools/esm_tools (last access 01.08.2024). The Matlab codes used for the analysis of the model output will be made available upon request to the corresponding author.

A processed minimum dataset of the FESOM data can be accessed under https://doi.org/10.5281/zenodo.15120056

The CCLM model is available from the CCLM community website: https://clmcom.scrollhelp.site/clm-community (last access: 07 August 2024). CCLM data are available in the DKRZ long-term archive: https://hdl.handle.net/21.14106/cb44f718061ed8e9cdeb574d51113b64ec781564 (last access: 07 August 2024, (Zentek and Heinemann, 2023)).

*Author contributions.* VT ran the FESOM simulations, conducted the analyses, and prepared the figures. CN and VT prepared the FESOM model code. GH and RZ prepared the CCLM model code and ran the simulations. VT, RT and TS contributed to the interpretation of the results. All authors contributed to the writing and editing of the manuscript.

*Competing interests.* The authors declare no competing interests.

*Acknowledgements.* This work was funded by Deutsche Forschungsgemeinschaft SPP 1158 under grants SE2901/2, HE2740/33 and TI296/9. CCLM simulations used resources of the Deutsches Klimarechenzentrum (DKRZ) granted by its Scientific Steering Committee (WLA) under project ID bb0958.

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
