# Peer review of "Simulated density reorganization on the Weddell Sea continental shelf sensitive to atmospheric forcing"

_EGUsphere, 2024_

## Referee Comment (RC1)

**Review of *Regime shift caused by accelerated density reorganisation on the Weddell Sea continental shelf with high-resolution atmospheric forcing* by Teske et al.**

The study uses a forced ocean-sea ice model to analyse the density structure on the continental shelf and near the continental shelf break in the Filchner Trough of the Weddell Sea. The manuscript presents present-day conditions as well as future conditions using atmospheric forcing from atmosphere models run under the SSP3-7.0 emission scenario. Two different atmospheric forcing datasets of varying horizontal resolution are used. The simulation with atmospheric forcing from a higher-resolution model simulates onshore transport of Warm Deep Water resulting in a shift of the continental shelf waters from cold to warm. This change in the water mass properties does not occur in the simulation forced with atmospheric forcing from a coarser atmospheric model. The manuscript highlights the sensitivity of the simulated projection of water mass properties in the study region to the applied atmospheric forcing.

The manuscript is overall-well structured, and the figures support the mechanisms described in the text. The data interpretation is robust, and the figures are clear and easy to read. My main comments are on the main effect of grid resolution for the atmospheric forcing product and the presentation of the results.

I detail below my main concerns followed by specific comments that are aimed to improve the manuscript.

**Main comments**

1) The manuscript highlights the role of high-resolution atmospheric forcing for the simulated ocean. The result is based on the comparison of the future scenario runs REF and FECO. The text mentions that the atmospheric forcing in FECO (from a higher-resolution regional model) e.g. results in warmer air temperatures. The manuscript does not discuss the following point: Is the higher resolution of the atmospheric model important because (i) it changes the mean climate (e.g. warmer air temperatures) and thereby the ocean or (ii) the small-scale variability itself changes the ocean? Or is the simulated atmosphere in the higher-resolution model different due to other aspects of the model configuration? The manuscript shows the sensitivity of the density structure to surface forcing but does to my understanding not explain why the high-resolution forcing is required. Another low-resolution model might simulate a different mean climate that could give similar results to FECO. Please elaborate.

2) I struggled multiple times to follow the argumentation as the text uses many long sentences and lacks summary sentences that guide the reader. The discussion and conclusion are an exception and much easier to follow. I encourage the author team to edit the other sections of the manuscript to enhance readability.
   - Introduction: lacks flow within the individual paragraphs
   - Method section: please add overview sentences that guide the reader. State the main observation first and then give more details. Also, the discussion and conclusion nicely highlight the relevance of the seasonality, but the point of the

seasonality is lost in the result section.

In the specific comment section below, I highlight some text passages, but the authors should edit the text beyond my comments below.

**Specific comments**

Title
- The title is very long and it is not clear what the message is. I encourage the authors to specify their main result of the paper. To me, the most interesting result is the sensitivity to the atmospheric forcing. I suggest to rewrite the title to something similar to "Simulated ocean density on the Weddell Sea continental shelf sensitive to applied atmospheric forcing"
- My main concerns with the current title are:
  *regime shift*: definition of regime is missing
  *accelerated density reorganisation*: accelerated compared with what? What is the density reorganisation?

Abstract

L3: I suggest replacing "characteristic V-shape in the density structure all along the continental slope" with "characteristic V-shape in the density structure **across** the continental slope"

L5/6: I suggest splitting the sentence into two: "...emission scenario**. The forcing is retrieved from** atmospheric model output from..."

L11: I suggest rewriting to "Using **forcing** data from an atmospheric model"

L11/12: "acceleration of the density redistribution" – not clear what this means, compared with what?

L12: First time the Filchner Trough is mentioned, it is not clear that this is the regional focus of the study.

L14: "grade of connectivity" – spatial connectivity? Please specify.

L16: "Our results also indicate..." this sentence repeats L11? This sentence is clearer to me, I suggest replacing the sentence in L11 with this one. Remove the "suggest" at the end of the sentence.

Introduction

L37: "southward direction" – here and elsewhere in the text southward/northward are used to describe directions. I suggest using "onshore" and "offshore" as the continental slope and cross-slope transects are often not meridional.

L40: "Variability" – temporal or spatial variability? Please clarify.

L42ff: The text says seasonality is important but the examples are for specific storm events and not seasons. The connection is not clear to me.

L48/49: I expect the overflow to be larger when there is less mixing with lighter surface waters / mixing with water that is not as light. Please clarify.

L49/50: This statement does not fit to the train of thought; the conundrum is not resolved: do we expect a sensitivity of DSW export to surface winds or not?

L57: Why is it important to use a global model? Is it because Antarctic meltwater has consequences for the global climate? Please elaborate.

L67: "regime shift": Please define the regime and what a shift of it means

L69-71: What did these studies find? What ocean processes changed in response to different atmospheric forcing?

Methods

L87: "on unstructured-mesh methods" – please reword, grammatically not a correct sentence

L89: Reword to "three-equation **parameterisation**"

L92: I suggest removing "via"

L98: I suggest rewriting to "The FESOM **REF simulation** is forced" to match the wording of the next paragraph.

L100: I suggest rewording to "component, **and which was developed** as a contribution to"

L199: I suggest rewording to "The follow **variables**"

L120-121: Why are the variables listed here? Do they differ to the low resolution forcing in REF?

L125: "V-shape" of what? V-shape of cross-slope isopycnals? Please specify (here and elsewhere in the manuscript).

L133-134: It is not clear to me what was done in step 2. Please split up into shorter sentences and elaborate.

L134/135: Move sentence on delta y_n below equation (1) where the other parts of the equation are described.

L141: "disruptions in longitudinal direction" – should this be in zonal direction?

Results
L152: I suggest rewording to "is found **approx. 500 m deeper** above…"

L155/156: "due to the deepening of the slope current along its path following the continental slope" – please provide evidence for the statement.

L165: Fig. S3 is referenced before Fig. S2. Please swap order of the two figures.

L165: Comment on the fact that the shallowest maximum depth is reached in autumn.

L167: How do the 50 m variation in depth compare to the grid cell thickness of the model at this depth? Is it more than one grid cell (which would mean the changes is not as drastic)?

L170/171: I suggest rewording to "in autumn **steepens** the **onshore** arm again"

L176: Please reword to "the horizontal temperature gradient **at** 300 m depth". Also: which direction does *horizontal* mean? Please specify.

L177: Give Fig. S4 reference earlier – it currently reads as if Fig S4 would show observations which is does not.

L178/179: Fig. 7 does not show Ekman downwelling, please provide evidence for the "weakened Ekman downwelling".

L187: I suggest rewording to "becomes **asymmetrical and shallower** towards the end"

L189: Please provide evidence for "a shoaling of the slope current over the course of the century"

L197: "enhances existing wind stress curl patterns in winter" – The change is hard to see in Fig 6a-d, it is possibly better to plot the anomaly instead

L200: Should all panels Fig. 6e-h (not just Fig. 6e) be referenced here?

L201: Please reword to "compared to the beginning of the century"

L201/202: "Regional variability is slightly reduced" – what aspect of Fig 6e-h shows this? What does "regional variability" mean in this context?

L204: "(not shown)" – Fig. 4 does show sea ice concentration changes.

L204/205: "a southward shift of the wind field increases areas of downwelling above the continental slope" – please provide evidence.

L205/207: I suggest rewording to "The impact of sea ice on the surface stress curl is especially visible in autumn **when** the Filchner Trough is covered approximately halfway by sea ice." Also please provide a figure reference for this statement.
L211: "long-term trend in up- and downwelling patterns", please give evidence/elaborate.

L215: Please provide a figure reference for this statement.

L217: "additional decrease" – additional to what?

L218: I suggest removing "of transition time" to streamline the text.

L218-220: Please provide figure reference.

L222: Please define "regime shift", e.g. "as seen by the much warmer shelf temperatures"

L223: "near bottom current across the sill" – please provide evidence

L224-226: This sentence is not clear to me and disturbs the flow of the text. Is the point here that REF shows increasingly stronger seasonal pulses, but they do not lead to a "regime shift"? Is this seen in the temperature field of Fig. 7? Please clarify.

L227-229: Doe we expect the historic warming trend to match the projected warming trend?

L227: "warming of the slope current" – here and at multiple other locations in the text, the term "slope current" is used to describe changes in the offshore part of the continental slope. Maybe a better wording can be used as the slope current itself is a dynamical feature and was never introduced in the manuscript by showing the velocity field in the model. I was quite confused every time the slope current was mentioned.

L232-233: move this information up to (near) the beginning of the paragraph. This is a main observation that is very useful to give early on so the reader can follow the argumentations of the paragraph, i.e. please state the obvious first.

L234: Please rephrase to "before the regime shift **occurs** in FECO"

L242-243: "Stronger Ekman downwelling in autumn, but a late onset of the freezing season" – I do not understand this sentence, please clarify.

L247: "reduction of density" – where is the density reduced? Please elaborate

L248: Please rephrase to "large in **the** Filchner Trough"

L252: "on the continental shelf" – is this the Filchner Trough? Please clarify.

L253: "in the slope current" – and this is the offshore region? Please clarify, see also my comment on L227.

L254: "reduced sea-ice formation" – What is the connection here to sea ice?

L252-253: Changes in temperature and salinity do not contribute linearly to changes in density. Is this incorporated in the assessment? I find this sentence difficult to understand, please clarify.

L257: "onset of the near-bottom current" – please provide evidence.

L258-259: "loss of the southern arm of the V-shape" – Does this mean it is a fresh shelf now?

L263: "the V-shape is formed also at a greater depth" – I do not see this in Fig. 8c, the V-shape in temperature or density? The isopycnals in Fig 8c are shallower, what exactly is deeper? Please clarify.

L264: "however, ..." Please start a new sentence here. The second half of the sentence is not clear as the term V-shape is used when a few sentences earlier the text says the onshore arm is lost. Perhaps rewrite to "isopycnals are much deeper in FECO than in REF". Or remove this part completely as the next sentence does not pick up on this point.

L265-266: Are the isopycnals deeper after the onset of the bottom current? Please clarify.

L271: "15-year": Which time period is this exactly? FECO simulates multiple time periods.

L271: "approx. 1000 m" – Where is the information of the 1000 m thermocline depth in the year 2000 shown? Am I meant to see this in Fig. 10?

L275-277: Please rewrite the sentence and split into two. Also, the statement is the same as in the first sentence of the paragraph? Please clarify. Also I expected now a discussion of the density ratio but that is missing or do I misunderstand?

L281: "Fig. 11" Please reference Fig. 11a

L282-286: I suggest simplifying this text, e.g., to "The southward mWDW (...) transport and the GOC are significantly correlated with a 3 month lag in REF (p=...,r=...) and in FECO (p=...,r=...). There is also a weak correlation between the outflowing DSW (...) and the GOC in REF (p=...,r=...) while such correlation is missing in FECO (p=...,r=...)."

L288-289: "decouple the weak correlation" – I do not understand this sentence, please clarify.

Figures

Figure 1:
- I suggest rewording "red line" to "red **rectangle**" (same for green line)
- I suggest rewording "Location of areas" to "Location of **subregions**"

Figure 6:
- First time that seasons are defined – please give definition in the main text when seasons are first mentioned.

Figure 7:
- "Colored lines show the relative temperature change…" is this the orange line? Why is the orange line explained again below? Could this be streamlined to "Colored lines show the relative change of the horizontally and vertically averaged temperature and salinity compared to the year 2000. Orange lines are for REF and red lines for FECO."

Figure S4:
- The red lines to highlight the -0.3°C and -0.7°C isotherms are difficult to see. Perhaps plotting them in black but bold improves readability.
- What does "zonal average" mean – only one (meridional) transect is shown here? Please clarify.

Figure S7:
- What is the "inset"? The map in panel d)? But that shows more than the continental shelf. Please clarify.

Figure S9:
- Please rephrase to "Linear regression"

---

## Author Comment (AC1)

**Authors' response to Review No 1**

Vanessa Teske*[1,2], Ralph Timmermann[1], Cara Nissen[3,4],
Rolf Zentek[5,6], Tido Semmler[1,7], Günther Heinemann[5]

[1]Alfred Wegener Institute for Polar and Marine Research, D-27570 Bremerhaven, Germany

[2]Department for Biogeochemical Modelling, GEOMAR, D-24148 Kiel, Germany

[3]Department of Atmospheric and Oceanic Sciences and Institute of Arctic and Alpine Research, University of Colorado, Boulder, Boulder, Colorado, USA

[4]Department of Freshwater and Marine Ecology, Institute for Biodiversity and Ecosystem Dynamics, University of Amsterdam, Netherlands

[5]Department of Environmental Meteorology, University of Trier, D-54286 Trier, Germany

[6]German Weather Service, D-63067 Offenbach, Germany

[7]Met Éireann, 65-67 Glasnevin Hill, D09 Y921 Dublin, Ireland

* Corresponding author: vanessa.kolatschek@awi.de

First, we would like to thank the reviewer for their helpful and detailed feedback on our manuscript. The additional questions and enquiries helped us to improve the quality of the text and further our understanding of the different processes and their connections.

In the following, we answer the reviewers' comments in turn in blue. New text added to the manuscript or modified from the original manuscript appears in *italics*.

**Main comments**

1. The manuscript highlights the role of high-resolution atmospheric forcing for the simulated ocean. The result is based on the comparison of the future scenario runs REF and FECO. The text mentions that the atmospheric forcing in FECO (from a higher-resolution regional model) e.g. results in warmer air temperatures. The manuscript does not discuss the following point: Is the higher resolution of the atmospheric model important because (i) it changes the mean climate (e.g. warmer air temperatures) and thereby the ocean or (ii) the small-scale variability itself changes the ocean? Or is the simulated atmosphere in the higher-resolution model different due to other aspects of the model configuration? The manuscript shows the sensitivity of the density structure to surface forcing but does to my understanding not explain why the high-resolution forcing is required. Another low-resolution model might simulate a different mean climate that could give similar results to FECO. Please elaborate.

   > This is a valid point. Within the scope of this study, we are unable to cleanly separate the effects of higher resolution in the forcing data and the different physics of the two atmospheric models. The aim of this study was to analyse the effect of different atmospheric forcings on a) intrusions of mWDW into Filchner Trough and b) the stability of the ASF. The effects are briefly discussed in the introduction and the discussion of the manuscript. We realize that we should have been more careful in how and when to use 'high-resolution' as a descriptor for the COSMO model data to highlight differences to the second forcing dataset. We thus went through the text and corrected the wording wherever it seemed useful. An additional paragraph has been added to the section "Discussion: Limitations and caveats" containing the following: *Based on the results in REF and FECO, the two atmospheric data sets impact the ocean in different ways. However, it is not straightforward to cleanly separate the effects of a higher atmospheric resolution and the different atmospheric states the two atmospheric models provide. It remains unclear whether the higher resolved model output from*

*CCLM impacts the ocean differently than the AWI-CM data because it simulates a different mean climate (e.g. warmer air temperatures) or because it resolves small-scale variability in the atmosphere. We note that the differences in the wind field between CCLM and AWI-CM can be explained by the improved representation of gradients in the orography of Antarctica (Mathiot et al., 2011, Elvidge et al., 2011; Cape et al., 2015), supporting, at least in part, the importance of resolving small-scale atmospheric variability.*

Additionally, we also hypothesised that resolved mesoscale atmospheric processes may intensify the seasonality of the V-shape, the on-shore mWDW transport and the export of DSW in the Weddell Sea. While we definitely see an increase in the on-shore transport of mWDW in FECO compared to REF (see Fig. 10d for 2086 to 2100), FECO exports on average a bit less DSW from the shelf than REF. These changes are consistent for the historical time period from 2000 to 2014 and the scenario simulations 2036-2050 and 2086-2100. FECO also produces changes in the seasonality of the V-shape compared to REF that change the geometry of the V-shape over the course of a year. We do not identify an increase in the seasonality itself (e.g. depth changes). These results have been added to the Results and are summarized in the Conclusion as follows: *We could show that the application of forcing data with resolved mesoscale atmospheric processes increases the on-shore mWDW transport as hypothesized. However, we observed a small decrease in the average DSW export from the shelf and no visible increase in the seasonality of the V-shaped density distribution above the continental slope.*

2. I struggled multiple times to follow the argumentation as the text uses many long sentences and lacks summary sentences that guide the reader. The discussion and conclusion are an exception and much easier to follow. I encourage the author team to edit the other sections of the manuscript to enhance readability.

   - Introduction: lacks flow within the individual paragraphs

   - Method section: please add overview sentences that guide the reader. State the main observation first and then give more details. Also, the discussion and

conclusion nicely highlight the relevance of the seasonality, but the point of the seasonality is lost in the result section. In the specific comment section below, I highlight some text passages, but the authors should edit the text beyond my comments below.

Thank you for the feedback. We reduced the length of sentences and added more structure to the text.

**Specific comments**

**Title**

- The title is very long and it is not clear what the message is. I encourage the authors to specify their main result of the paper. To me, the most interesting result is the sensitivity to the atmospheric forcing. I suggest to rewrite the title to something similar to "Simulated ocean density on the Weddell Sea continental shelf sensitive to applied atmospheric forcing"
- My main concerns with the current title are:

*regime shift:* definition of regime is missing

*accelerated density reorganisation:* accelerated compared with what? What is the density reorganisation?

We acknowledge that regime and regime shift are not defined in the title. As it might not easily recognisable what a regime shift in this environment includes, we decided on the new title: *"Simulated density reorganization on the Weddell Sea continental shelf sensitive to atmospheric forcing"*

**Abstract**

**L3:** I suggest replacing "characteristic V-shape in the density structure all along the continental slope" with "characteristic V-shape in the density structure across the continental slope"

Done.

**L5/6:** I suggest splitting the sentence into two: "...emission scenario. The forcing is retrieved from atmospheric model output from..."

*Done.*

**L11:** I suggest rewriting to "Using forcing data from an atmospheric model"

*Done.*

**L11/12:** "acceleration of the density redistribution" – not clear what this means, compared with what?

*The density changes on the shelf are happening faster in the model runs forced with COSMO output in comparison to the simulation forced with ECHAM data. To clarify, we added:"…compared to the simulation forced with ECHAM output."*

**L12:** First time the Filchner Trough is mentioned, it is not clear that this is the regional focus of the study.

*We reworded the previous sentence to include: "…into the Filchner Trough in the southern Weddell Sea…"*

**L14:** "grade of connectivity" – spatial connectivity? Please specify.

*It is spatial connectivity. We rephrased the sentence to:"…we define a spatial grade of connectivity to …"*

**L16:** "Our results also indicate…" this sentence repeats L11? This sentence is clearer to me, I suggest replacing the sentence in L11 with this one. Remove the "suggest" at the end of the sentence.

*We rephrased the paragraph and moved the mentioned sentence further up.*

**Introduction**

**L37:** "southward direction" – here and elsewhere in the text southward/northward are used to describe directions. I suggest using "onshore" and "offshore" as the continental slope and cross-slope transects are often not meridional.

*Very nice suggestion. This has been changed throughout the text.*

**L40:** "Variability" – temporal or spatial variability? Please clarify.

*In this case it concerns temporal variability, caused by changes in both DSW export and mWDW import. It has been changed to: Temporal variability of the V-shape has been linked to …".*

**L42:** The text says seasonality is important but the examples are for specific storm events and not seasons. The connection is not clear to me.

*The V-shape is sensitive to changes in the wind field. A storm event is just a more extreme example. We changed the sentence around to first describe sensitivity to winds and then seasonality (which is still wind related):* "The V-shape has been shown to be sensitive to the wind field (Graham et al., 2013). [...] Additionally, the V-shape shows seasonal variability in depth that has been associated with variations in the along-shore wind strength (Graham et al., 2013)."

**L48/49:** I expect the overflow to be larger when there is less mixing with lighter surface waters / mixing with water that is not as light. Please clarify.

*The overflow transport is geostrophically controlled. Lighter water masses at greater depth enhance the density gradient across the shelf break and therefore the overflow transport. To clarify, we added:* "... geostrophically controlled..."

**L49/50:** This statement does not fit to the train of thought; the conundrum is not resolved: do we expect a sensitivity of DSW export to surface winds or not?

*Based on published literature, the community does not seem to agree on whether or to what extent the DSW export is sensitive to the surface winds. Previous studies (listed in the text) suggest yes, however Stewart & Thompson (2015a) suggest only a 20 % variability in response to wind. We added to the text:* However, existing studies do not all agree on the importance of winds. [...] In contrast, three-dimensional eddy-resolving simulations by Stewart & Thompson (2015a) showed only low sensitivity of the DSW export to wind strength.

**L57:** Why is it important to use a global model? Is it because Antarctic meltwater has consequences for the global climate? Please elaborate.

*It is important to use a global model to include remote atmospheric and oceanic connections. These influences can not be incorporated adequately within a regional model. We added the following to the introduction:* On longer timescales, the ASF is remotely influenced by large-scale climate modes such as the Southern Annular Mode (SAM) and the El Niño-Southern Oscillation (ENSO; Armitage et al., 2018, Spence et al., 2014, 2017). *The discussion was expanded by the following:* Armitage et al. (2018) found that the southward contraction of Westerlies during positive SAM enhances northward Ekman transport, dropping coastal sea level and weakening the ASF. In negative SAM phases the opposite happens. While

*the impact of long-term positive SAM trends on the ASF are uncertain, modelling studies suggest that a drop in Antarctic coastal sea level could weaken the ASF, leading to shoaling of the isopycnal on the continental slope (Spence et al., 2014, 2017). Similar events take place during El Niño events, where an anticyclonic atmospheric pressure anomaly over the Amundsen and Bellinghausen Seas lead to a weakening of the ASF in the Pacific Sector of the Southern Ocean (Armitage et al., 2018, Spence et al., 2014). ENSO should show their influence in REF and FECO in the same manner, because its origin in equatorial latitudes is not part of the CCLM domain and therefore any influence they have does not differ between REF and FECO. While the analysis of remote connections is not included in this study, it should be considered in the future.*

**L67:** "regime shift": Please define the regime and what a shift of it means.

The regime shift changes the main water mass in the Filchner Trough from cold DSW to warm mWDW by increasing mWDW inflow. We specified:*"...from a DSW-dominated trough circulation to a mWDW-dominated circulation in the Filchner Trough."*

**L69-71:** What did these studies find? What ocean processes changed in response to different atmospheric forcing?

Consequences include flooding of the ice shelf cavity with warm water from the deep ocean, a rise in basal melt rates of the ice shelves, reduced density of the exported shelf waters a, less efficient deep-ocean carbon and oxygen transfer. The studies cited in the text all produce a regime shift in the Filchner Trough before the end of the 21st century. The following has been added to the text: *Previous studies have described a possible regime shift in the Filchner Trough from a DSW-dominated trough circulation to a mWDW-dominated circulation in the trough and possible consequences, including flooding of the ice shelf cavity with warm water from the deep ocean, a rise in basal melt rates of the ice shelves, reduced density of the exported shelf waters and a less efficient deep-ocean carbon and oxygen transfer (Hellmer et al., 2012, Timmermann et al., 2013, Naughten et al., 2021, Nissen et al., 2022, 2023, 2024).*

**Methods**

**L87:** "on unstructured-mesh methods" – please reword, grammatically not a correct sentence.

> The sentence has been rephrased to:*"FESOM is a global ocean general circulation model with an unstructured mesh..."*

**L89:** Reword to "three-equation parametrisation"

> Done.

**L92:** I suggest removing "via"

> Done.

**L98:** I suggest rewriting to "The FESOM REF simulation is forced" to match the wording of the next paragraph.

> We kept these sentences as they were because we would like to keep the distinction of REF as the reference simulation, but we removed the brackets.

**L100:** I suggest rewording to "component, and which was developed as a contribution to"

> Instead, we added the following to split the sentence:*"The data was created..."*

**L199:** I suggest rewording to "The follow variables"

> Changed to:*"The following atmospheric variables..."*

**L120-121:** Why are the variables listed here? Do they differ to the low resolution forcing in REF?

> The forcing variables are the same for REF and FECO. We changed the paragraph to make this clear: *The following atmospheric variables are used to drive all simulations: 2m-temperature, 10m-wind, downward longwave and shortwave radiation at the ocean surface, mean sea level pressure, 2m specific humidity and total precipitation.*

**L125:** "V-shape" of what? V-shape of cross-slope isopycnals? Please specify (here and elsewhere in the manuscript).

> The V-shape describes indeed the shape of the cross-slope isopycnals. We added this specification at the first mentioning of the V-shape in each subsection.

**L133-134:** It is not clear to me what was done in step 2. Please split up into shorter sentences and elaborate.

The paragraph has been changed to: As a second step, we find the horizontal density minimum at the chosen depth of 250 m (in the example in Fig, **??**a it is $27.5\,kg/m^3$). This is repeated for each meridional grid coordinate, creating a number of density minima (red dots in Fig. **??**b).

**L134/135:** Move sentence on delta y_n below equation (1) where the other parts of the equation are described.

Done

**L141:** "disruptions in longitudinal direction" – should this be in zonal direction?

It has been changed.

**Results**

**L152:** I suggest rewording to "is found approx. 500 m deeper above. . ."

Done.

**L155/156:** "due to the deepening of the slope current along its path following the continental slope" – please provide evidence for the statement.

It is visible in the downward shift of the warm core just at the off-shore edge of the profiles in Fig. 3. We reworded to: *"...due to the deepening of the warm core of the mWDW transported by the slope current along its path along the continental slope (Fig. 3).*

**L165:** fig. S3 is referenced before fig. S2. Please swap order of the two figures.

Done.

**L165:** Comment on the fact that the shallowest maximum depth is reached in autumn.

The depth of the isopycnals reacts with a lag to changes in the sea ice formation, but more immediately to changes in the wind field. The following text has been added to the paragraph: *The minimum sea ice extent is reached in general in February. While near-coastal sea ice formation starts again in March, the increased export of DSW from the continental shelf is delayed due to the distances involved between the area of sea ice formation along the coasts and the continental slope (see also Fig. S6e-h). The isopycnals at the continental slope therefore only deepen later, leading to the shallowest maximum depth being reached in autumn.*

**L167:** How do the 50 m variation in depth compare to the grid cell thickness of the

model at this depth? Is it more than one grid cell (which would mean the changes is not as drastic)?

> Between 600 and 750 m, layer thickness increases from 20.6 to 21.7 m. An average depth difference of 50 m covers 2 to 3 mesh layers, making the change not irrelevant. We added: *"Such a vertical distance corresponds to two to three layers in the ocean grid."*

**L170/171:** I suggest rewording to "in autumn steepens the onshore arm again"

> Done.

**L176:** Please reword to "the horizontal temperature gradient at 300 m depth". Also: which direction does horizontal mean? Please specify.

> This has been changed to: *"the meridional temperature gradient at 300 m depth (Fig. S4)."*

**L177:** Give fig. S4 reference earlier – it currently reads as if fig S4 would show observations which is does not.

> Done.

**L178/179:** fig. 7 does not show Ekman downwelling, please provide evidence for the "weakened Ekman downwelling".

> This is a typo and should have said Fig. 6 with positive surface stress curl being equivalent to Ekman downwelling on the southern hemisphere. The error has been corrected.

**L187:** I suggest rewording to "becomes asymmetrical and shallower towards the end"

> Done.

**L189:** Please provide evidence for "a shoaling of the slope current over the course of the century"

> This is visible in the uplift of the thermocline, of which we see the last 15 years of REF in Fig. 10a. A timeseries of the depth of the thermocline for the whole REF simulation has been added to the Supplementary material and is now referenced in the text (Fig. S11).

**L197:** "enhances existing wind stress curl patterns in winter" – The change is hard to see in fig 6a-d, it is possibly better to plot the anomaly instead

> Instead of plotting the anomaly, we adjusted the axis limits for the panels a) to

d) which makes differentiating between the two curves much easier.

**L200:** Should all panels fig. 6e-h (not just fig. 6e) be referenced here?

Yes. It has been changed.

**L201:** Please reword to "compared to the beginning of the century"

Done.

**L201/202:** "Regional variability is slightly reduced" – what aspect of fig 6e-h shows this? What does "regional variability" mean in this context?

Regional variability means the frequent changes of in strength of the up- and downwelling following the profile from south to north, particularly in winter and spring (Fig. 6e-f). In summer, the distribution is much smoother, resembling the surface wind stress in summer (panel c). The sentence has been reworded: *Variability along the profile is slightly reduced so that the surface stress curl resembles the wind stress curl more closely at the end of the century (compare Fig. 6c/g).*

**L204:** "(not shown)" – fig. 4 does show sea ice concentration changes.

We only show sea ice formation rates and sea ice thickness, but no concentration.

**L204/205:** "a southward shift of the wind field increases areas of downwelling above the continental slope" – please provide evidence.

This should have said *"northward shift of the wind field"* and has been changed accordingly. A figure of the difference in surface stress curl between winter and summer has been added to the supplements.

**L205/207:** I suggest rewording to "The impact of sea ice on the surface stress curl is especially visible in autumn when the Filchner Trough is covered approximately halfway by sea ice." Also please provide a figure reference for this statement.

The text has been adjusted to: *The impact of sea ice on the surface stress curl is especially visible in autumn when the Filchner Trough is partly covered by sea ice with a northward decreasing thickness (Fig. S8d).*

**L211:** "long-term trend in up- and downwelling patterns", please give evidence/elaborate.

In this sentence, we wanted to describe overarching patterns in up-and downwelling instead of long-term trend. The text has been adjusted to: ... *we conclude that the up-and downwelling patterns are created by the sea ice distribution rather than the wind field.*

**L215:** Please provide a figure reference for this statement.

In principle, the increased on-shore transport of mWDW can be seen in Fig. 11. However, this is not convenient for the flow of the manuscript. We therefore changed the sentence to: *The application of the regional higher-resolved CCLM forcing in FECO affects the heat transport onto the continental shelf (Fig. 7c)*

**L217:** "additional decrease" – additional to what?

Additional was meant in relation to the decreasing salinity already happening in REF due to the changing climate and reducing sea ice formation. Because this does not seem to be clear, the sentence has been reworded to: *Reduced freezing rates along the coasts in winter and higher melt rates in summer (Fig. S3) due to higher air temperatures (not shown) compared to REF lead to a decrease of the mean salinity in the Filchner Trough by up to 0.2 psu over the course of 10 years.*

**L218:** I suggest removing "of transition time" to streamline the text.

Done.

**L218-220:** Please provide figure reference.

An additional figure showing the mean wind speed and mean wind speed anomaly between FECO and REF in a 5-year mean (2010-2014) has been added as Supplementary Fig. S9.

**L222:** Please define "regime shift", e.g. "as seen by the much warmer shelf temperatures"

The suggestion has been added.

**L223:** "near bottom current across the sill" – please provide evidence

It is visible in FECO (Fig. 8d), where the bottom layers show high temperatures, as well as warm water filling the Filchner Trough from the bottom. The reference has been added to the text. Additionally, we provide an animated gif of monthly mean bottom temperatures in the southern Weddell Sea. The warm current entering the Filchner Trough is first prominently visible in 2093.

**L224-226:** This sentence is not clear to me and disturbs the flow of the text. Is the point here that REF shows increasingly stronger seasonal pulses, but they do not lead to a "regime shift"? Is this seen in the temperature field of fig. 7? Please clarify.

REF shows increasingly stronger seasonal pulses but they do not lead to a regime

shift before 2100. FECO also shows these pulses, but here we find a regime shift happening in 2093. This can be seen in Fig.7. To clarify, the whole paragraph has been reworked.

**L227-229:** Do we expect the historic warming trend to match the projected warming trend?

Yes and no. At the beginning of the simulation, it should match the historical warming trend, because there is no reason it should experience a sudden change. That it does this, adds confidence to the model simulation. It is possible that the warming trend strengthens or slows down, but as we do not explore the mechanisms behind the warming of the slope current and ACC, it is difficult to evaluate at this stage. No changes to the text were applied.

**L227:** "warming of the slope current" – here and at multiple other locations in the text, the term "slope current" is used to describe changes in the offshore part of the continental slope. Maybe a better wording can be used as the slope current itself is a dynamical feature and was never introduced in the manuscript by showing the velocity field in the model. I was quite confused every time the slope current was mentioned.

The slope current transports mWDW. In this context it is often easier to refer to the temperature of the current instead of the temperature of the mWDW transported by the current. We went through the manuscript to correct this oversight and added an introduction to the Antarctic Slope Current in the introduction.

**L232-233:** move this information up to (near) the beginning of the paragraph. This is a main observation that is very useful to give early on so the reader can follow the argumentations of the paragraph, i.e. please state the obvious first.

The whole paragraph has been restructured and split up. Previous comments have been taken into account.

**L234:** Please rephrase to "before the regime shift occurs in FECO"

Done.

**L242-243:** "Stronger Ekman downwelling in autumn, but a late onset of the freezing season" – I do not understand this sentence, please clarify.

Ekman downwelling is an aspect dictating the characteristics of the depth of the V-shape. With strong Ekman downwelling in autumn, the isopycnals deepen

again. However due to higher atmospheric temperatures, sea ice formation and in particular dense water formation start later. The isopycnals lose the characteristic V-shape because the continental shelf is filled with less dense water than later in winter. This creates a situation similar to fresh-shelf regions like at the coast of Dronning Maud Land, consisting of only the off-shore arm of the V-shape, with the isopycnals intersecting with the continental slope. We added the following clarifications in the text: *Stronger Ekman downwelling in autumn, but a late onset of the freezing season lead to a situation that temporarily resembles fresh-shelf regions like the Dronning Maud Land section (Fig. **??**a) where the V-shape is missing its on-shore arm and the isopycnals cancel at the slope.*

**L247:** "reduction of density" – where is the density reduced? Please elaborate.

*in the Filchner Trough* has been added to the text.

**L248:** Please rephrase to "large in the Filchner Trough"

Done.

**L252:** "on the continental shelf" – is this the Filchner Trough? Please clarify.

Yes, it is and has been changed to "in the trough"

**L253:** "in the slope current" – and this is the offshore region? Please clarify, see also my comment on L227.

In this instant, we describe the mWDW transported along the continental slope by the slope current. This has been specified: *…while the trend in the mWDW transported by the ASC is driven by a combination of warming and freshening (Fig. S11).*

**L254:** "reduced sea-ice formation" – What is the connection here to sea ice?

As has been explained further above, reduced freezing rates along the coast in winter and higher melt rates in summer in FECO compared to REF lead to freshening of the shelf waters. And while this is indeed a prominent factor, changes in the temperature and salinity of the mWDW transported by the slope current also play a role in shaping the density distribution and development across the Filchner Trough sill. We added the information the the paragraph to make the connection clearer: *Separately assessing the trends in temperature and salinity reveals that the density trend in the trough is driven by freshening due to reduced*

*freezing rates along coasts in winter and higher melt rates in summer, ...*

**L252-253:** Changes in temperature and salinity do not contribute linearly to changes in density. Is this incorporated in the assessment? I find this sentence difficult to understand, please clarify.

While temperature and salinity do not contribute linearly to changes in density, plotting the mean temperatures and salinity values in a T-S diagram (Fig. S1) shows that in this case, salinity changes, in particular in the trough, are the dominating force behind the density changes. We amended the paragraph as follows: *The change in density is larger in FECO in the Filchner Trough than at the off-shore location. From this, we conclude that reduced sea-ice formation is not the only factor influencing the density distribution across the continental slope, but the dominating one for the existence of V-shaped cross-slope isopycnals.*

[Figure]

Figure S1: T-S-Diagram of annual mean potential temperature and salinity over time in REF (a, c) and FECO (b, d) in the Filchner Trough (a, b) and at the Filchner Trough sill depth off-shore (c, d). The arrows indicate the shift over time. Isopycnals relative to 1000 kg/m3 are shown in black.

**L257:** "onset of the near-bottom current" – please provide evidence.

The bottom current can be seen in Fig. 8d in the 2096-2100 mean temperature. This current starts to become visible in 2093. Also we refer to answer to comment on line 223. We changed the text to: *Visible as a sudden increase in the average temperature in the Filchner Trough (Fig. 7c), and as a layer of warm water at the bottom of the Filchner Trough (Fig. 8d), the inflow of mWDW in FECO in 2093*

**L258-259:** "loss of the southern arm of the V-shape" – Does this mean it is a fresh shelf now?

In contrast to a fresh shelf, the Filchner Trough starts to fill with mWDW, creating a distribution that reminds more of the warm shelves in the Amundsen Sea. Because we have no profile of the Amundsen Sea for comparison, no changes were applied to the text.

**L263:** "the V-shape is formed also at a greater depth" – I do not see this in fig. 8c, the V-shape in temperature or density? The isopycnals in fig 8c are shallower, what exactly is deeper? Please clarify.

he V-shape is actually flatter/shallower. However the on-shore arm of the 27.4-isopycnal is pronounced, while in FECO, it is not. The main difference on the shelf is basically the range of density on the shelf with (as described) stronger stratification in FECO. The text has been amended to: *This has the effect of increasing the stratification of the water column. In combination with low sea-ice formation rates and reduced mixing (not shown) during the freezing season, the seasonal variations in the southern arm of the V-shape vanish at depths below approx. 450 m (Fig. 8d). In REF, the density distribution on the continental shelf leads to the formation of the typical V-shape(Fig. 8c). The isopycnals above the continental shelf also experience stronger seasonal depth variations. Seasonal variations of the depth and position of the V-shape and the position of the northern arm can be found in FECO and REF. The on-shore arm of the V-shape in FECO decreases strongly in vertical extent. [...] The weaker stratification in REF has the effect of making the on-shore arm shallower than 450 m more pronounced. Below this depth, the isopycnals take on a similar shape as in FECO.*

**L264:** "however, . . . " Please start a new sentence here. The second half of the sentence is not clear as the term V-shape is used when a few sentences earlier the text says the

onshore arm is lost. Perhaps rewrite to "isopycnals are much deeper in FECO than in REF". Or remove this part completely as the next sentence does not pick up on this point.

*We refer to the answer and changes explained above.*

**L265-266:** Are the isopycnals deeper after the onset of the bottom current? Please clarify.

*This part describes the relative height of the on-shore shape of the isopycnals, but not its depth. Due to the bottom of the Filchner Trough being filled with mWDW from off-shore, the isopycnals become more levelled. We changed the text to: From a height difference in spring of approx. 200 m between the deepest point of the V-shape and the shallowest point above the continental shelf, the 27.4 kg m$^{-3}$ isopycnal position reduces its vertical extend to a range of approx. 80 m after the bottom current onset.*

**L271:** "15-year": Which time period is this exactly? FECO simulates multiple time periods.

*This describes the last period from 2086 to 2100 (where the interesting stuff is happening). We changed the text to: In FECO, the thermocline remains above the Filchner Trough sill depth throughout the FECO simulation from 2086 to 2100...*

**L271:** "approx. 1000 m" – Where is the information of the 1000 m thermocline depth in the year 2000 shown? Am I meant to see this in fig. 10?

*The year 2000 is indeed not shown in Fig. 10. It has been added as Fig. S12 to the supplements and referenced in the text.*

**L275-277:** Please rewrite the sentence and split into two. Also, the statement is the same as in the first sentence of the paragraph? Please clarify. Also I expected now a discussion of the density ratio but that is missing or do I misunderstand?

*The time series of the density ratio between the maximum density in Filchner Trough and the density at the Filchner Trough sill does not tip in favour of the the sill density in REF at all. In FECO this happens multiple times and over a longer period of time. These times agree well with times of increased on-shore heat transport when considering the slight delay caused by the time it takes the mWDW to flow into the trough. The paragraph has been extended to: This agrees*

*with the conclusions presented by Haid et al. (2023) suggesting that another key factor for or against an inflow of mWDW onto the continental shelf in addition to the depth of the mWDW at the slope is the density ratio between the dense water on the shelf and the mWDW of the ASC (see also Nissen et al., 2023). FECO only experiences the regime shift after the density ratio between the DSW of the Filchner Trough and the mWDW in the ASC changes in favour of the mWDW (Fig. 12b). And while the thermocline in REF does also cross sill depth during the last five years of the simulation, the density ratio remains in favour of the DSW, therefore preventing the intrusion of large amounts of mWDW into Filchner Trough.*

**L281:** "fig. 11" Please reference fig. 11a

Done.

**L282-286:** I suggest simplifying this text, e.g., to "The southward mWDW (. . . ) transport and the GOC are significantly correlated with a 3 month lag in REF (p=. . . ,r=. . . ) and in FECO (p=. . . ,r=. . . ). There is also a weak correlation between the outflowing DSW (. . . ) and the GOC in REF (p=. . . ,r=. . . ) while such correlation is missing in FECO (p=. . . ,r=. . . )."

We modified the text to be easier to follow and included the suggestion :*After 2093, the southward mWDW ($\Theta$ >-0.8°C) transport (Fig. 12d)) and the GOC are significantly correlated with a 3 month lag in REF (p=7e-7, r=0.36) and in FECO (p=0.0005, r=0.35). There is also a weak correlation between the outflowing DSW ($\Theta$ <-0.8°C; Fig. 12d) and the GOC in REF (p=0.0001, r=0.29) while such correlation is missing in FECO (p=0.27, r=0.08).*

**L288-289:** "decouple the weak correlation" – I do not understand this sentence, please clarify.

It was meant to highlight that the changes in the density of the DSW in the Filchner Trough are influencing the V-shape and with that the calculated GOC from the beginning of the simulation. The reduced density of the DSW leads to a change in the export behaviour and how or if it disturbs the V-shape. We replaced *decouple* with *remove*.

**figures**

**figure 1:**

- I suggest rewording "red line" to "red rectangle" (same for green line)

- I suggest rewording "Location of areas" to "Location of subregions"

> Done.

**figure 6:**

- first time that seasons are defined – please give definition in the main text when seasons are first mentioned.

> Definitions of seasons have been added to subsection "Present-day seasonality of the V-shape at Filchner Trough (REF simulation)"

**figure 7:**

- "Colored lines show the relative temperature change…" is this the orange line? Why is the orange line explained again below? Could this be streamlined to "Colored lines show the relative change of the horizontally and vertically averaged temperature and salinity compared to the year 2000. Orange lines are for REF and red lines for FECO."

> Yes, this makes the caption more compact and has been changed.

**figure S4:**

- The red lines to highlight the -0.3°C and -0.7°C isotherms are difficult to see. Perhaps plotting them in black but bold improves readability.

- What does "zonal average" mean – only one (meridional) transect is shown here? Please clarify.

> Zonal average is correct, however the figure does not show the temperature at 300 m depth along the Filchner Trough Profile but averaged over the Filchner Trough Sill area as shown in Fig. 1. This was corrected. The colours have been changed.

**figure S7:**

- What is the "inset"? The map in panel d)? But that shows more than the continental shelf. Please clarify.

> We meant the map added to panel d). The area used for calculation lies within the red polygon. As this does not seem to be prominent enough, we added the polygon in a more prominent colour and replaced *(see inset)* with *outlined in red*

*in the map* in the figure caption.

**figure S9:**

- Please rephrase to "Linear regression"

Done.

---

## Author Comment (AC2)

**Authors' response to Review No 2**

Vanessa Teske*[1,2], Ralph Timmermann[1], Cara Nissen[3,4],

Rolf Zentek[5,6], Tido Semmler[1,7], Günther Heinemann[5]

[1]Alfred Wegener Institute for Polar and Marine Research, D-27570 Bremerhaven, Germany

[2]Department for Biogeochemical Modelling, GEOMAR, D-24148 Kiel, Germany

[3]Department of Atmospheric and Oceanic Sciences and Institute of Arctic and Alpine Research, University of Colorado, Boulder, Boulder, Colorado, USA

[4]Department of Freshwater and Marine Ecology, Institute for Biodiversity and Ecosystem Dynamics, University of Amsterdam, Netherlands

[5]Department of Environmental Meteorology, University of Trier, D-54286 Trier, Germany

[6]German Weather Service, D-63067 Offenbach, Germany

[7]Met Éireann, 65-67 Glasnevin Hill, D09 Y921 Dublin, Ireland

* Corresponding author: vanessa.kolatschek@awi.de

First, we would like to thank the reviewer for their helpful and detailed feedback on our manuscript. The additional questions and enquiries helped us to improve the quality of the text and further our understanding of the different processes and their connections. In the following, we answer the reviewers' comments in turn in blue. New text added to the manuscript or modified from the original manuscript appears in *italics*.

**Main comments**

I have 2 main comments on the V shape metrics that could bring more robustness to the manuscript:

1. What are the robustness of your results to the choices made in your definition? Is it possible, based on your algorithm to define the maximum depth of the V-shape (or the thickness of the V shape) by applying it to multiple depth? This could support some discussion points of your manuscript and give a more complete understanding of the variability of the V shape structure.

   The algorithm is best applied in depths were the V-shape of the isopycnals is not intersected by the continental shelf. At the sill of the Filchner Trough, this is the case down to approx. 600 m, less in the surroundings. To avoid significant interactions with the mixed layer near the surface, we also apply the algorithm only below 200 m. Between 200 and 600 m, the GOC is robust against depth selection, which is not the case above and below this depth interval. As can be seen in Fig. C1, the upper 200 m show distinct seasonal variability in GOC. Below 200 m, the algorithm detects events with low GOC over multiple model layers, even though the strength of detected events varies (Fig. C1b). While the algorithm can also be applied to deeper layers, this is not insightful without additional information on the existence of the on-shore arm of the V-shape. The algorithm has no criterium for the existence of the V-shape, but instead searches for the density minimum in a plane, explaining why we restrict its application to between 200 and 600 m where we know the V-shape exists. Given the definition of the GOC, it is not possible to define a maximum depth of the V-shape based on the GOC at multiple depths. To achieve this, an additional criterium is needed to test if the minimum was found at a boundary node. If that is the case, the isopycnals do not have the characteristic V-shape. By adding this second parameter (V-shape: yes/no), the GOC could give a more comprehensive understanding of the dynamic and stability of the V-shape. We amended the following paragraph in the discussion of the GOC: *The introduction of*

*the GOC provides a metric for the stability of the ASF in areas of cross-slope transport in dense-shelf areas. However, not all features of the V-shape of the density distribution are included in its definition. Because it searches for density minima at a chosen depths, inferring the existence of the complete V-shaped distribution is not possible from the GOC alone. This information needs to be compiled before applying the algorithm. The GOC is useful to detect strong changes in the cross-slope density distribution that remove the dip in the isopycnals completely or lead to a lateral displacement of the V-shape in a section parallel to the continental slope. However, the GOC is not able to differentiate between a lot of small displacements that just exceed the distance threshold $d_L$ and few, large displacements. The selection of $d_L$ as a maximum displacement threshold also influences the result. A distance larger than ten times the grid size reduces the number of recognised events by 12.5% compared to a $d_L$ of two times the grid size; with $d_L$ 100 times the grid size, the number of recognised events decreases by 29.1%. Additionally, the GOC is sensitive to the chosen depth. Generally, the GOC is robust against depth selection within the water column below the mixed layer but above the depth of the continental shelf to prevent an intersection of the isopycnals with the sea floor within the V-shape. To minimize the impact of seasonal variability in the upper ocean and to avoid an intersection of the chosen depth layer with the continental slope within the main feature of the V-shape, a depth of $250\,m$ was chosen for this study.*

2. About the processes, based on your introduction the present day V-shape structure is mostly driven by the cascading of the DSW. After the regime shift, you mention that there is still a V-shape structure but at shallower depth and without DSW cascading. What is the processes that maintained the V-shape without the entrainment of the overlying surface water to the descending flow? And can you put that in relation with region that already experimented a regime shift like the Amundsen Sea and as far as I know, don't have a V-shape (even weak) in the range you mentioned here for 2100.

   The continued presence of the on-shore arm of the V-shaped density distri-

[Figure]

Figure C1: GOC in REF at different layer depths.

bution in the upper layers even after the regime shift in FECO is a result of two processes. The sea ice formation in winter, while strongly reduced compared to the beginning of the century, leads to the formation of DSW. This DSW has a lower density at the end of the century, but is exported from the continental shelf (Fig. 11c of the manuscript). In the upper ocean, the DSW on the continental shelf presents a barrier for the Ekman transport against which surface water is pushed and subducted. The difference to the beginning of the simulation/21st century lies in the depth to which the surface water is subducted. While the transport of mWDW creates, at first glance, a picture similar to that of a warm shelf of Amundsen Sea (Thompson et al., 2018), the presence of the V-shape in the density of the upper layers presents a distinct difference. Atmospheric wind patterns like the Amundsen Sea Low or the Bellinghausen Sea Low create a more variable surface wind forcing (Turner et al., 2013), removing the mechanism for the formation of

the off-shore arm of the V-shape. Additionally, the warm shelf regions are not areas of DSW formation, therefore removing the other mechanism that leads to the formation of the distinct V-shape in the density distribution at the continental slope. We added the following text to the discussion to highlight these aspects: *The warming of the Filchner Trough through increased heat transport creates a warm shelf that, in some parts, resembles the warm shelf in the Amundsen Sea (Thompson et al. 2018). The exception is the continued presence of the V-shape in the density distribution of the upper ocean at Filchner Trough. Atmospheric patterns like the Amundsen Sea Low create a more variable surface wind (Turner et al., 2013), and the absence of DSW formation on the shelves of the Amundsen Sea remove the mechanisms for the formation of the V-shaped density distribution.*

**Specific comments**

**Section 1**

**l76-l83:** I think, 'section' instead of 'chapter' should be used.

Done.

**l76-l83:** You described well what you will present in section 3.2, 3.3, 3.4 and 3.5. It is worth adding a line on section 2 and section 4 for completeness.

The paragraph has been extended: *A brief description of the models and the methods used for analysis are given in Section 2, all results will be discussed in Section 4.*

**Section 2**

**Section 2.1**

**l86-l96:** what is the bathymetry source? What eddy parametrisation are you using? What bulk formulation for the atmospheric flux?

To answer these questions, the following has been added to the method section of the manuscript:*The model utilizes the ocean bathymetry, ice shelf geometry and grounding line position from RTopo-2 (Schaffer et al. 2016). Parametrizations of*

*subgrid-scale fluxes use the Gent and McWilliams (1990) scheme and Redi (1982) rotated tracer diffusion. Further detail on parameterizations in FESOM-1.4 has been provided by Wang et al. (2014). Bulk formulae for momentum transfer between the atmosphere and the ocean/sea-ice surface are quadratic functions of the velocity difference. The drag coefficient and the transfer coefficients for latent and sensible heat fluxes vary as a function of stability following Large and Yeager (2004).*

**l90-l91:** It is not clear to me the resolution. You mention 4km around ice shelf cavities and 25km at 75S. It is unclear to me because there is cavities around and northward than 75S.

The numbers are specifically given for the Weddell Sea. We specified this in the text as follows: *The variable horizontal resolution of the ocean mesh ranges from (minimum) 4 km around Antarctica and its adjacent ice shelf cavities, via 25 km at 75°S in the open ocean in the Weddell Sea, to 250 km at the equator.*

**l93:** can you mention the vertical resolution range? For example at the surface and 1000m depth.

The vertical resolution decreases with depth. While the grid has a spacing of the layers of 5 m in the upper 100 m of the water column, at 1000 m the layer are separated by 30 m. The layer thickness increases up to 337.5 m at nearly 6000 m depth. The information has been added to the text: *In the vertical, the mesh has 99 depth levels (z-levels) of varying increasing thickness with depth, spanning between 5 m near the surface, up to 337.5 m at nearly 6000 m depth (Gurses et al., 2019; Nissen et al., 2023).*

**Section 2.2**

**l110:** Do you know the possible impact of atmospheric forcing produced by model in 'forecast mode'? Could it impact the solution?

Due to the short simulation times of 12 hours spin-up and one day before reinitialization, the CCLM results remain close to the forcing data set of AWI-CM output which is used to constrain the model at its outer boundaries. It is impossible for CCLM to drift far away from its forcing, and tendencies arising in these short

simulation chunks cannot accumulate. Whether this is a strength or a weakness of the concept may be debatable; we see it as a strength because it emphasizes local and resolution effects rather than larger-scale model physics differences. Initial adjustment processes at each of the restarts, on the other hand, are removed by ignoring the first 12 hours of CCLM simulation.

**l110:** A brief description of the performance of CCLM on representing present day Antarctica climate and comparison with the two well know Antarctic regional model RACMO and MAR is welcome.

In response to this comment, the following paragraph has been added to the manuscript: *CCLM was evaluated for the present days climate for the Weddell Sea region in Zentek and Heinemann (2020) using near-surface data, upper-air data, ERA reanalyses (ERA-Interim and ERA5) and the Antarctic Mesoscale Prediction System (AMPS, Powers et al. 2012) for the period 2002–2016. CCLM showed a good representation of temperature and wind for the Weddell Sea region.*

**Section 2.3**

**l133:** Could you explain why 250m and not another depth? How robust are your conclusion to a change in this choice? In your figure 2a, depending of what depth I choose, I can have variation of about 0.5 degree in latitude. If we look at fig. 2b, most of your points are with a 0.5 degree latitude band. Therefore, I am wondering the robustness of the result about meridional change/deplacement of the front.

The depth of 250 m was chosen based on the depth of the continental shelf and to reduce the influence of seasonally changing surface water. The chosen plane does not interface with the ocean floor and cross-sections show that the V-shape exists at this depth, which might not be the case at greater depths (see also explanation above). While the position of the V-shape can vary, the absolute position is not part of the calculation for the GOC, only the relative distances between neighbouring density minima. A meridional shift of the complete V-shape does therefore not influence the result of the GOC. This information has been added to the text: *Because the algorithm only includes the relative distance between the density minima, the GOC is robust against horizontal displacement of*

[Figure]

Figure C2: Hovmöller diagram of the monthly mean density relative to $1000\,kg/m^3$ in the wider Filchner Trough area (see map) in REF. The white line shows the mixed layer depth.

*the whole V-shaped density structure.*

**l133:** Is 250m an issue with the winter mixed layer? Is the 250 m depth line within or below the mixed layer depth?

*The winter mixed layer depth is underestimated by the model (Fig. C2). As a result, 250 m is below the mixed layer for the majority of the time. If the GOC was affected by the winter mixed layer depth, we would expect a clear seasonal decrease in the value, which is only the case for the upper approx. 200 m of the water column (Fig. C1a). Additionally, we found only weak or no correlation between the export of DSW from the shelf an the GOC, indicating that also increased export in winter or spring does not have a strong impact on the GOC.*

**no specific line:** Based on your algorithm, if you applied it at each depth level, could you define the vertical extent of the V-shape and latitude variability in depth of the V shape position to complete your analysis and add robustness to some affirmation.

*At greater depth, the isopycnal intersects the continental slope, and the on-shore arm of the V-shape is (probably) missing. The GOC can't differentiate between a minimum in a V and a minimum of a sloping plane. It its current form, the GOC can not be used to define the vertical extent of the V-shape (see also response to main comment).*

**Section 3**

**Section 3.1**

**obs comparison:** I would like to see a discussion on the realism of the shelf properties, V-shape and offshore properties with respect to observations on the present period for REF and FECO simulations. This will bring more confidence in your results.

> The circulation patterns on and off the continental shelf are well represented (see also Fig. S1 in the Supplements) by the model. The salinity of the shelf water is generally lower in the model results than in observations. The temperature on the continental shelf is up to 0.6°C warmer in the model results than in observations. The largest differences in the temperature between model and observations can be found in the area of the highly variable mWDW current entering the continental shelf. The following comparison to observational data has been added to the manuscript: *Generally, the circulation in REF on and off the continental shelf are well represented by the model. Comparison to observations from mooring and CTD data (Schröder, 2010; Schröder et al., 2014, 2016, 2019; Janout et al., 2019) show that the salinity on the continental shelf is slightly underestimated by the model results by up to 0.15 psu (Fig. S13b), however some areas in particular in Filchner Trough can overestimate the salinity slightly. The temperature on the on the shelf is up to 1°C warmer in the model than in observations (Fig. S13a), but largest differences in the temperature can be found at the eastern slope of the Filchner Trough where the highly variable mWDW current flows onto the continental shelf. At the continental slope, the temperature difference can be larger, while the mWDW transported by the ASC is colder by up to 0.5°C. FECO produces a similar temperature distribution as REF (Fig. S14a), but produces lower salinity values (Fig. S14b).*

**l159:** I found the sentence hard to follow when looking at Fig. S1a as this is not a temperature map.

> The figure refers to the transport patterns on the shelf, not the temperature distribution. The warm water filling the Larsen Ice Shelf cavity can also be seen in Fig. 3d. We added a figure reference and reworded the sentence to: *In addition,*

*mWDW at temperatures around -1°C can be found on the continental Shelf and in the Larsen Ice Shelf cavity (Fig. 3c, d). This water mass originates from Ronne Trough and follows the coast northward (Fig. S1a).*

**Section 3.2**

**l166:** Fig. S3 called before S2.

Done.

**l168:** About the 'smaller amplitude', can you give a number?

The thermocline can change depth by up to $100\,m$ over the course of one year according to Hattermann et al. (2018). The paragraph has been amended as follows to include this information: *The seasonality is consistent with observed variations in the thermocline depth at the Filchner Trough sill, though the variation is low compared to the amplitude of over $100\,m$ observed (Hattermann, 2018).*

**Figure 3:** Can you make it bigger? Maybe in a 2x2 panel with grid lines?

We adjusted the figure.

**l177:** There is no observations in Fig. S4. Reformulate.

Done.

**l178:** Clarify what you call DSW export in Fig. S6 because Fig. S6 show meridional and zonal velocity without indication of density. Furthermore, right south of the cross, there is a wall of one pixel. This let suggest, that what you call DSW export is not the southward transport of the bottom cell. You should add a clear definition of what you call DSW and make it more visible in Fig. S6 and in all the other figure you discuss DSW.

We selected the maximum meridional velocity as export velocity at the indicated location, because comparison between Fig. S6 and Fig. 4 of the main manuscript shows the dense water in this position. For clarification, we added some isopycnals to Fig. S6 and corrected some numbers.

**l179:** I found hard to understand the 'weakened Ekman downwelling' with your temperature and salinity hovmuller in Figure 7. Can you plot a more direct variable like the Ekman pumping for example?

Positive surface stress curl corresponds to Ekman downwelling on the southern

hemisphere. This is visible in Fig. 6. Fig. 7 was wrongly referenced and has been corrected.

**Section 3.3**

**Fig. 5 and 4:** Bigger (maybe 2x2?) and with the same size (I have the feeling that when you added the legend it shrink the panel vertically in Fig. 5 wrt Fig 4).

Done.

**l191:** What isopycnal are you using to affirm that. Is it still valid if you used another isopycnal?

As can be seen in Fig. 4 and Fig. 5, we look at the 27.7 and 27.55 isopycnals because they lie at similar depths and (nearly) don't interface the surface. The statement is also true for other isopycnals.

**Fig. 5:** Why do you color a different isopycnal in Fig. 5 compared to Fig. 3 and 4? Maybe also homogenise the color. For exemple the 27.8 is black in Fig3 and white in Fig4?

The 27.7 isopycnal does not show the V-shape at the end of the century. Instead it intersects with the continental slope, similar to the 27.8 isopycnal. The placement (2x2) and colours have been adjusted.

**l190:** You mention that the depth of the V shape is reduced by using the 27.8 isopycnal. Why using 27.8 to compute the depth of the V shape when in 2100 time slice, there is no left arm and so no V shape with this isopycnal?

We used the 27.8 isopycnal because we chose a density that does not intersect the surface to isolate our analysis somewhat from changes in the mixed layer. It is also not a characteristic of this particular isopycnal, but rather of the whole V-shape. Additionally, even though at the end of the century the 27.8 isopycnal terminates at the continental slope, vertical movement propagates through large parts of the water column, and the isopycnal needs to be consistent to compare the beginning and the end of the century. It is possible that we underestimate the movement by choosing an isopycnal that is not part of the 'real' V-shape at the end of the century, but this should not affect our results.

**l200:** Can you clarify the 80%. On Fig S7b, a reduction of 80% should lead to $40\,\mathrm{cm}$

[Figure]

Figure C3: Change in sea ice thickness between the 15-year means of 2086-2100 and 2014-2000 in REF.

thickness in 2100 (2 m in 2000).

The time series shown in Fig. S7 shows the annual average sea ice thickness over the continental shelf of the southern Weddell Sea. However, thickness varies greatly between the Ronne and Filchner sections, with much greater sea ice thickness being reached in the Ronne sector of the continental shelf. The thicker the ice, the more can melt. To be more precise, the winter sea ice reduces at least by 35% between the 15-year mean of 2000 to 2014 and the 15-year mean 2086 to 2100 (Fig. C3). We have clarified this in the text: *With a reduction of winter sea-ice thickness by at least 35% - in large areas even more - compared to the beginning of the century, some of the up - and downwelling areas are redistributed.*

**Section 3.4**

**Fig. S8:** add grid on the figure

Done.

**Section 3.6**

**l263:** I don't really understand. Can you reformulate the bloc discussing Fig. 8c. It is not clear what to look at.

The on-shore arm of the V-shape in FECO is only very weakly pronounced after the mWDW bottom-current onset (Fig. 8d). The isopycnals above the continental

shelf also show no seasonal variation in depth or form. In REF, the density distribution on the continental shelf leads to the formation of the typical V-shape (Fig. 8c). The isopycnals above the continental shelf also experience stronger seasonal depth variations, which might be caused by their shallower position closer or in the mixed surface layer. The paragraph in the manuscript has been reworked as follows: *Visible as a sudden increase in the average temperature in the Filchner Trough (Fig. ??c), and as a layer of warm water at the bottom of the Filchner Trough (Fig. 8d), the inflow of mWDW in FECO in 2093 brings the bottom density in the Filchner Trough closer to that of the ASC. This has the effect of increasing the stratification of the water column. In combination with low sea-ice formation rates and reduced mixing (not shown) during the freezing season, the seasonal variations in the southern arm of the V-shape vanish at depths below approx. 450 m (Fig. ??d). In REF, the density distribution on the continental shelf leads to the formation of the typical V-shape (Fig. ??c). The isopycnals above the continental shelf also experience stronger seasonal depth variations due to the location closer or in the mixed layer. Seasonal variations of the depth and position of the V-shape and the position of the northern arm can be found in FECO and REF. The on-shore arm of the V-shape in FECO decreases strongly in vertical extent. From a height difference in spring of approx. 200 m between the deepest point of the V-shape and the shallowest point above the continental shelf, the $27.4\,kg\,m^{-3}$ isopycnal position reduces its vertical extend to a range of approx. 80 m after the bottom current onset.*

**l270:** When you are discussing thermocline depth, it is probably worth it defining exactly what you mean. Is it the base of the thermocline, the top, the middle, a specific isotherm depth?

Thermocline is defined as the depth of greatest vertical temperature gradient. It has been added to the text. *In the following, the thermocline is defined as the greatest vertical temperature gradient.*

**Fig. 10:** Define the thermocline depth. Probably worth merging Fig. 10 and 11 to be able to compare easily timing between curves. Add also vertical grid to ease even more the comparison.

> The thermocline depth has been calculated as the depth at which the greatest vertical temperature gradient can be found along the continental slope. We combined the two figures into one.

**l287:** What is the explanation for the presence of a V-shape after a regime shift or during intrusion of mCDW on the shelf. In the introduction, the presence of DSW cascading need critical in the formation of the V-shape. Once the mCDW flood the shelf, there is no more cascading. So I am surprised by the presence of a V shape in this case.

> The V-shape is constrained to the upper ocean. At the depth of the inflow, no V-shape is formed. The V-shape in the upper ocean still follows the same principles as in the beginning of the simulation, however the effects are smaller and the mixing does not go as deep any more. An explanation has been added to the text (see main comments).

**Section 4**

**Section 4.2**

**l344:** Same question as for section 3.6 on the processes that drive the V shape without DSW cascading.

> See above and main comment.

**l355:** please reformulate 'Ryan (2017) showed that during ... suppress the isopycnals ...' sentence. I don't really understand the first part. What is suppressing the isopycnals at the continental slope.

> Maybe suppress is the wrong choice, subduct works better. The production of DSW and its export across the shelf break in winter entrain more overlying water masses. This reduces or stops the on-shore transport of mWDW. The text has ben changed accordingly.

**l358:** Please reformulate 'in shallower than sill depth'.

> Done.

**section 4.4**

**l378:** add a description of the eddy param in model description.

A brief description of the model sub-scale parametrization has been added to the methods (see comment on lines l86-l96).

**l382:** You mentioned issue with the lack of eddy. Is it also true in your case with the eddy parametrisation ?

We haven't tested it, but it might hypothetically be possible that with the reduced eddy presence due to the lack of resolution, the model might underestimate cross-slope volume transport and the V-shape. It is possible but would need further testing.

**Conclusion**

**l404:** Please reformulate, I had to re-read it multiple time to understand it. I found some term vague like 'temporary disturbance'.

The sentence has been split for easier readability. *The criterion of spatial coherency of the V-shape along the continental slope, quantified by the grade of connectivity, is not usable as a tool to predict an imminent regime shift because the onshore transport of modified Warm Deep Water is, contrary to our expectations, not a result of a weakening of the slope front. Instead, the cross-slope transport leads to a temporary disturbance of the ASF and the associated V-shape. While the density minimum is completely restored after disruption in present-day climate, in a warming climate the distinct V-shape remains confined to the upper ocean.*